# Continuous muscle, glial, epithelial, neuronal, and hemocyte cell lines for *Drosophila* research

**Nikki Coleman-Gosser[1†], Yanhui Hu[2†], Shiva Raghuvanshi[1†], Shane Stitzinger[1†], Weihang Chen[2], Arthur Luhur[3], Daniel Mariyappa[3], Molly Josifov[1], Andrew Zelhof[3], Stephanie E Mohr[2], Norbert Perrimon[2,4]\*, Amanda Simcox[1,5]\***

[1]Department of Molecular Genetics, Ohio State University, Columbus, United States; [2]Drosophila RNAi Screening Center and Department of Genetics, Harvard Medical School, Boston, United States; [3]Drosophila Genomics Resource Center and Department of Biology, Indiana University, Bloomington, United States; [4]Howard Hughes Medical Institute, Chevy Chase, United States; [5]National Science Foundation, Alexandria, United States

**Abstract** Expression of activated Ras, Ras$^{V12}$, provides *Drosophila* cultured cells with a proliferation and survival advantage that simplifies the generation of continuous cell lines. Here, we used lineage-restricted Ras$^{V12}$ expression to generate continuous cell lines of muscle, glial, and epithelial cell type. Additionally, cell lines with neuronal and hemocyte characteristics were isolated by cloning from cell cultures established with broad Ras$^{V12}$ expression. Differentiation with the hormone ecdysone caused maturation of cells from mesoderm lines into active muscle tissue and enhanced dendritic features in neuronal-like lines. Transcriptome analysis showed expression of key cell-type-specific genes and the expected alignment with single-cell sequencing and in situ data. Overall, the technique has produced in vitro cell models with characteristics of glia, epithelium, muscle, nerve, and hemocyte. The cells and associated data are available from the *Drosophila* Genomic Resource Center.

\*For correspondence:
perrimon@genetics.med.
harvard.edu (NP);
simcox.1@osu.edu (AS)

†These authors contributed
equally to this work

**Competing interest:** The authors declare that no competing interests exist.

## Editor's evaluation

This valuable work describes the establishment and characterization of new cell lines derived from specific tissues of the fruit fly *Drosophila*. The evidence supporting the claims of the authors is convincing, with rigorous characterization of the cell lines and incorporation of their transcriptomes into *Drosophila* Gene Expression Tool website for user-friendly access. These lines will be a valuable resource that complements in vivo *Drosophila* genetics, improving biochemistry and facilitating high-throughput screening.

## Introduction

The use of cell cultures has been important for studying biological processes that are not easily accessible in whole organisms (***Klein et al., 2022***). A number of advances in mammalian cell cultures, for instance, development of 3D/organoid cultures (***Rossi et al., 2018***), improved genome editing tools to manipulate induced pluripotent stem cells (***Hockemeyer and Jaenisch, 2016***), and better optimized media formulations for recombinant protein expression ***Ritacco et al., 2018*** have further enhanced the utility of mammalian cell culture systems. These advances are accompanied by the availability of several distinct mammalian cell lines derived from different tissue types. Similarly, the use of

**eLife digest** Fruit flies are widely used in the life and biomedical sciences as models of animal biology. They are small in size and easy to care for in a laboratory, making them ideal for studying how the body works. There are, however, some experiments that are difficult to perform on whole flies and it would be advantageous to use populations of fruit fly cells grown in the laboratory – known as cell cultures – instead.

Unlike studies in humans and other mammals, which – for ethical and practical reasons –heavily rely on cell cultures, few studies have used fruit fly cell cultures. Recent work has shown that having an always active version of a gene called *Ras* in fruit fly cells helps the cells to survive and grow in cultures, making it simpler to generate new fruit fly cell lines compared with traditional methods. However, the methods used to express activated *Ras* result in cell lines that can be a mixture of many different types of cell, which limits how useful they are for research.

Here, Coleman-Gosser, Hu, Raghuvanshi, Stitzinger et al. aimed to use *Ras* to generate a collection of cell lines from specific types of fruit fly cells in the muscle, nervous system, blood and other parts of the body. The experiments show that selectively expressing activated *Ras* in an individual type of cell enables them to outcompete other cells in culture to generate a cell line consisting only of the cell type of interest.

The new cell lines offer models for experiments that more closely reflect their counterparts in flies. For example, the team were able to recapitulate how fly muscles develop by treating one of the cell lines with a hormone called ecdysone, which triggered the cells to mature into active muscle cells that spontaneously contract and relax.

In the future, the new cell lines could be used for various experiments including high throughput genetic screening or testing the effects of new drugs and other compounds. The method used in this work may also be used by other researchers to generate more fruit fly cell lines.

insect cell lines also complements whole organismal studies and helped to illuminate many aspects of insect cell biology (*Luhur et al., 2019*) including development (*Sato and Siomi, 2020*), immunity (*Goodman et al., 2021*; *Chen et al., 2021*), host–pathogen relationships (*Smagghe et al., 2009*), in addition to biotechnological applications (*Hong et al., 2022*).

Fruit fly (*Drosophila melanogaster*) cell cultures are among the most widely used invertebrate cell cultures (*Luhur et al., 2019*). *Drosophila* cell lines are relatively homogenous, and highly scalable for both biochemical and high-throughput functional genomic analyses (*Debec et al., 2016*, *Baum and Cherbas, 2008*; *Zirin et al., 2022*; *Mohr, 2014*; *Viswanatha et al., 2019*). These features underlie their status as an important workhorse for scientific discovery in organismal development and as models for human disease. There are approximately 250 distinct *Drosophila* cell lines housed by the *Drosophila* Genomics Resource Center (DGRC) (*Luhur et al., 2019*). The majority of these cell lines, initially established by independent laboratories worldwide, were donated to the DGRC. A subset of 25 of these lines was subjected to transcriptome analysis, with the results demonstrating that approximately half of the transcripts expressed by each of these lines were unique such that even cell lines derived from the same tissue had distinct transcriptomic profiles (*Cherbas et al., 2011*). Furthermore, the transcriptional profiles of several imaginal disc lines analyzed were found to match profiles of cells from distinct spatial locations in the respective discs (*Cherbas et al., 2011*). All lines exhibited transcript profiles indicative of cell growth and cell division, and not cellular differentiation, as expected for proliferating cells (*Cherbas et al., 2011*). Thus, the transcriptional profiles of several *Drosophila* cell lines provided a platform for subsequent analyses. For instance, a few examples of the impact of this work include research into better understanding crosstalk between signaling pathways (*Ammeux et al., 2016*), exploring transcription factor networks (*Rhee et al., 2014*), establishing small RNA diversity (*Wen et al., 2014*), characterizing signaling pathways (*Neal et al., 2019*), nucleosomal organization (*Martin et al., 2017*) among multiple other utilities reviewed extensively (*Cherbas and Gong, 2014*; *Luhur et al., 2019*).

Over two-thirds of the *D. melanogaster* cell lines listed in the DGRC catalog were derived from whole embryos and the remainder are from various larval imaginal discs, the larval central nervous system, larval hemocytes, or adult ovaries. The potential of cells from these different sources to

**Table 1.** Cell lines analyzed.

| Tissue-type alignment | Genotype | Lines analyzed* | DGRC stock name and number | RRID |
|---|---|---|---|---|
| Glial | Repo-Gal4; Ras$^{V12}$; brat$^{dsRNA}$ | Rbr6 (parental)<br>Rbr6-2<br>Rbr6-4<br>Rbr6-F9 | repo>Ras bratdsRNA-L6, 282<br>repo>Ras bratdsRNA-L6-Clone2, 326<br>repo>Ras bratdsRNA-L6-Clone4, 327<br>repo>Ras bratdsRNA-L6-CloneF9, 328 | RRID:CVCL_XF57<br>RRID:CVCL_C7G9<br>RRID:CVCL_C7GA<br>RRID:CVCL_C7GB |
| Epithelial | btl-Gal4; UAS-P35; UAS-Ras$^{V12}$ | Btl3 (parental) | btl>Ras attP-L3, 332 | RRID:CVCL_B3N7 |
| | btl-Gal4; UAS-P35; attP, UAS-Ras$^{V12}$ | Btl7 (parental)<br>Btl8 (parental) | btl>Ras attP-L7, 285<br>btl>Ras attP-L8, 286 | RRID:CVCL_XF53<br>RRID:CVCL_XF54 |
| Muscle | 24B-Gal4; attP, UAS-Ras$^{V12}$ | 24B5 (parental)<br>24B5-B8<br>24B5-D8 | 24B>Ras attP-L5, 284<br>24B>Ras attP-L5-CloneB8, 323 | RRID:CVCL_XF52<br>RRID:CVCL_C7G6 |
| | 24B-Gal4; UAS-GFP; attP, UAS-Ras$^{V12}$ | 24BG1 (parental)<br>24BG1-F3[†]<br>24BG1-G1[†] | 24B>Ras attP GFP-L1, 283<br>24B>Ras attP-G1-CloneF3, 325<br>24B>Ras attP-G1-CloneG1, 324 | RRID:CVCL_XF51<br>RRID:CVCL_C7G8<br>RRID:CVCL_C7G7 |
| Neuronal | Act5C-GeneSwitch-Gal4; UAS-GFP; attP, UAS-Ras$^{V12}$ | ActGSB-6[‡]<br>ActGSI-2 | Act5C-GS>Ras attP-LB-Clone6, 329<br>Act5C-GS>Ras attP-GFP-LI-Clone2, 330 | RRID:CVCL_C7GC<br>RRID:CVCL_C7GD |
| Blood | Act5C-GeneSwitch-Gal4; UAS-GFP; attP, UAS-Ras$^{V12}$ | ActGSI-3 | Act5C-GS>Ras attP-GFP-LI-Clone3, 331 | RRID:CVCL_C7GE |

*Clones unless indicated.

[†]Do not differentiate into active muscle.

[‡]These cells do not express GFP, the reason for this is not known.

differentiate into adult cell types is not known. However, temporal transcriptional profiling of the Ecdysone response of 41 cell lines (*Stoiber et al., 2016*) provided evidence that cell lines exhibited varying levels of ecdysone sensitivity and potential for cellular differentiation, suggesting the possibility of developing cell-type-specific cell lines with the capacity to differentiate.

As well as having unknown cellular origins, most *Drosophila* cell lines arose spontaneously, and the time needed to develop a continuous cell line was often protracted. In contrast, expression of activated Ras, Ras$^{V12}$, using the Gal4-UAS system, resulted in the rapid and reproducible generation of continuous cell lines from primary embryonic cultures (*Simcox et al., 2008b*). The Ras method was used to develop an array of mutant cell lines by using appropriate genotypes to establish the primary cultures (*Simcox et al., 2008a*, *Lee et al., 2015*; *Kahn et al., 2014*; *Lim et al., 2016*; *Nakato et al., 2019*). To date all lines have been generated using ubiquitous expression of *UAS-Ras* with *Act5C-Gal4* and therefore the cell type in a given line is unknown.

Here, we describe a second-generation version of the Ras method in which Ras$^{V12}$ expression is restricted to a lineage by using tissue-specific Gal4 drivers. This genetic 'dissection' provides only the targeted cells with the survival and proliferation advantage conferred by Ras$^{V12}$ expression (*Simcox et al., 2008b*). As we show, the approach has been successful and resulted in the generation of cell lines with glial, epithelial, and muscle characteristics. Lines generated by broad Ras$^{V12}$ expression should also include those of specific cell types and by using single-cell cloning and cell type characterization (marker gene expression and RNAseq) we identified lines with neuronal and hemocyte characteristics. Collectively, these cell lines provide in vitro models for five different cell types and are expected to be a valuable resource for high-throughput and biochemical approaches, which rely on large numbers of homogeneous cells.

## Results

Primary cultures were established from embryos in which *UAS-Ras$^{V12}$* expression was restricted to glial, tracheal epithelial, and mesodermal cells using lineage-specific Gal4 drivers (*Table 1*, *Supplementary file 1*). A subset of continuous cell lines derived from each type of primary culture was analyzed with regard to cell morphology, the presence of proteins characteristic of specific cell types, and other attributes (*Table 1*, *Supplementary file 1*, *Supplementary file 2*; *Figure 1*). We also analyzed lines with neuronal- or hemocyte-like characteristics that were cloned from parental lines resulting from ubiquitous expression of *UAS-Ras$^{V12}$* (*Table 1*, *Supplementary file 1*, *Supplementary file 2*; *Figure 1*).

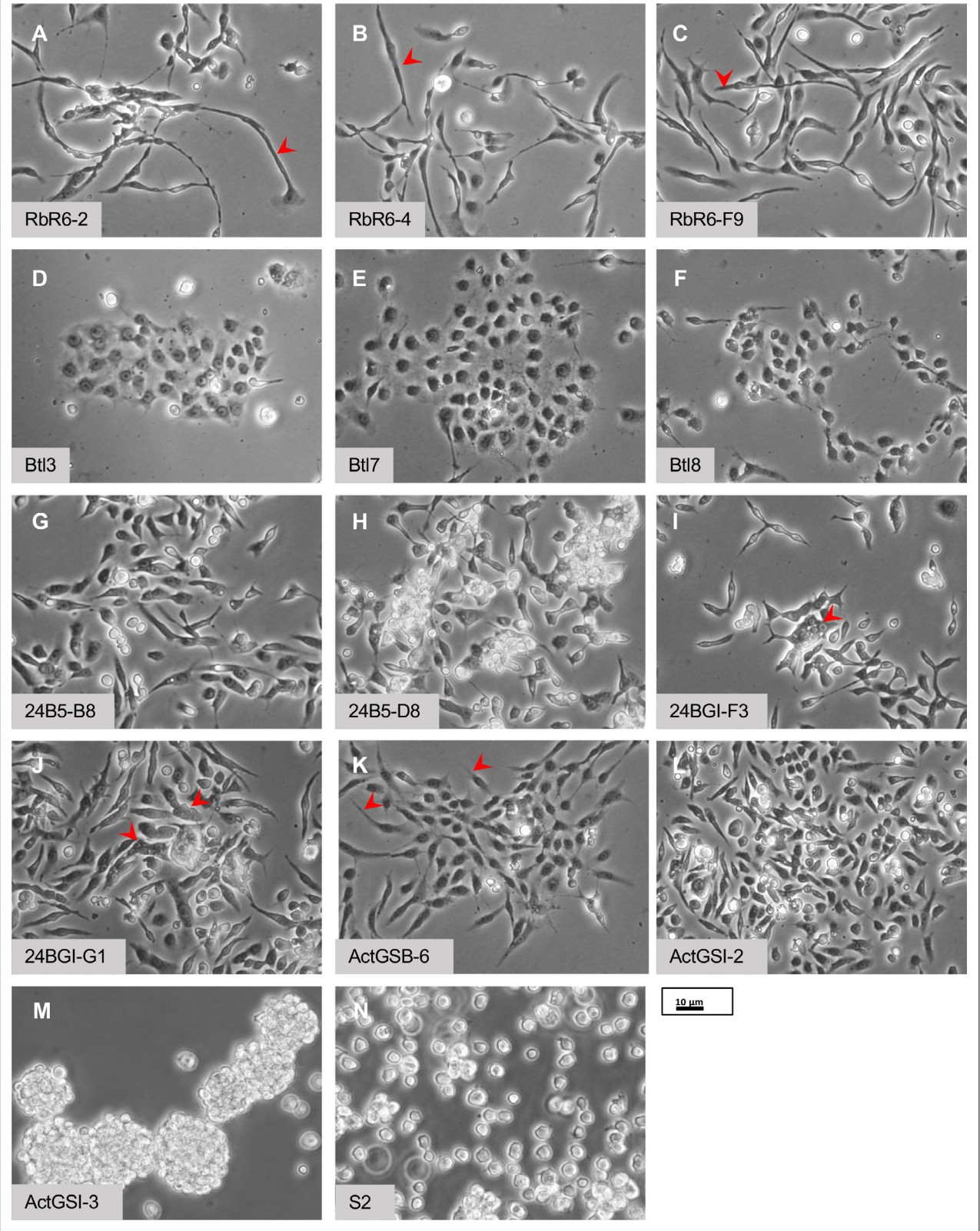

**Figure 1.** Morphology of cells. (**A–C**) Glial-lineage clones. The cells have an elongated morphology with variable lengths from approximately 20 to >50 µm (red arrowheads). (**D–F**) Tracheal-lineage cells. Btl3 and Btl7 cells form squamous epithelial sheets. Btl8 are closely associated but do not abut each other to form a sheet. (**G–J**) Mesodermal-lineage cells. The cells have a bipolar morphology. Multinucleate cells are frequently found in 24BGI-F3 and 24BG1-GI clones (red arrowheads). (**K, L**) Neuronal-like clones. ActGSB-6 cells are mainly bipolar; however, some have asymmetric processes or thin

*Figure 1 continued on next page*

*Figure 1 continued*

processes (red arrowheads). ActGSI-2 are bipolar. (**M**) Hemocyte-like clone ActGSI-3. The cells form floating clusters that increase in cell number as they proliferate. Individual cells have a round morphology. (**N**) Schneider's S2 cells. The cells are thought to be of hemocyte type and grow as single round cells in suspension. Scale bar = 10 μm.

We further analyzed the cell lines by RNAseq to determine the transcriptome and signaling pathways (*Figure 2* and *Figure 2—figure supplements 1–3*). The gene expression values (Fragments Per Kilobase per Million mapped fragments, FKPM) are provided in *Supplementary file 3*. The dataset (Ras cell lines) has been imported into the *Drosophila* Gene Expression Tool (DGET) database (https://www.flyrnai.org/tools/dget/web/), which is the bulk RNAseq data portal at *Drosophila* RNAi Screening Center (DRSC) (*Hu et al., 2017*). The TM4 package was used for making the plot in *Figure 2* (*Wang et al., 2017*). As expected, the transcriptomes of the new cell lines are distinct from those of existing cell lines (*Cherbas et al., 2011*; *Figure 2—figure supplement 1*) and new cell lines derived from the same Gal4 driver cluster with one another (*Figure 2—figure supplement 2*). Moreover, comparison of differentially expressed (DE) genes with RNAseq data from single-cell RNAseq data (*Li et al., 2022*; *Table 2*) or with known cell type-associated transcription factors (*Figure 2—figure supplement 3*) reveals that these cells express genes characteristic of specific cell types. The results of our detailed characterization are described according to cell type in the sections below.

## Glial-lineage cell lines

Repo is expressed exclusively in glial cells (*Xiong et al., 1994*). A *repo-Gal4* driver that recapitulates Repo expression was used to express *UAS-Ras$^{V12}$* (*Ogienko et al., 2020*; *Sepp et al., 2001*). This led to robust production of primary cultures however these failed to survive beyond early passages (*Supplementary file 1*). To counter potential cell death or modulate growth signaling, additional genotypes were tested including co-expression of *UAS-transgenes* encoding the P35 baculovirus cell survival factor, dsRNAs targeting tumor suppressors, or the Gal4 inhibitor Gal80$^{ts}$ (*Supplementary file 1*). Co-expression of a *UAS-brat$^{dsRNA}$* or expression of *tub-Gal80$^{ts}$* each produced a single line of cells that could be propagated for extended passages however the latter line was difficult to maintain and eventually lost (*Supplementary file 1*). The *repo-Gal4: UAS-brat$^{dsRNA}$; UAS-Ras$^{V12}$* (Rbr6) line has been passaged more than 50 times. The parental Rbr6 line and three clonal derivatives (Rbr6-2, Rbr6-4, and Rbr6-F9) have an elongated morphology and stained positive for Repo (*Table 1*; *Figures 1 and 3*, and *Figure 3—figure supplement 1*). A few cells expressed neuronal markers (*Figure 3—figure supplement 1*; *Supplementary file 2*). To induce differentiation, we gave cells two 24 hr ecdysone treatments separated by 24 hr to approximate the pulses of ecdysone during the larval to pupal transition. Cells from each of the clones survived treatment with ecdysone suggesting they are of adult type, two clones showed morphological changes and formed a network, and all continued to express Repo (*Figure 3* and *Figure 3—figure supplement 2*).

The results of RNAseq analysis revealed that the three Rbr6 clones have very similar expression patterns (*Figure 2—figure supplement 2*). In addition, their DE gene signatures are also a close match to gene signatures of glial cells as identified by single-cell RNAseq (*Table 2*) and to glial-associated genes reported in the literature. For example, *zydeco* (*zyd*), which encodes a potassium-dependent sodium/calcium exchanger, is upregulated in all three clones, consistent with the literature (*Zwarts et al., 2015*; *Featherstone, 2011*), and *gcm2*, a transcription factor, is upregulated in two clones (*Figure 2—figure supplement 3*). These data suggest the Rbr6 clones will be a useful in vitro source of glial cells.

## Tracheal epithelium-lineage cell lines

Breathless is expressed in the tracheal epithelium and a *btl-Gal4* driver was used to express *UAS-Ras$^{V12}$* (*Shiga et al., 1996*). Patches of cells with epithelial morphology proliferated in primary cultures and several continuous lines were generated (*Table 1*, *Supplementary file 1*). We were unable to derive clones of these using dilution or selection methods, which were successful for other cell types. Correspondingly, three parental lines were examined: Btl3, Btl7, and Btl8 (*Table 1*). All showed expression of the epithelial marker Shotgun/E-Cadherin (Shg/Ecad) and two grew in a squamous epithelial sheet with Ecad expression at the cell periphery (*Figures 1 and 4*, and *Figure 4—figure supplement 1*). In comparison S2 did not show peripheral expression of Ecad (*Figure 4*). Treatment of

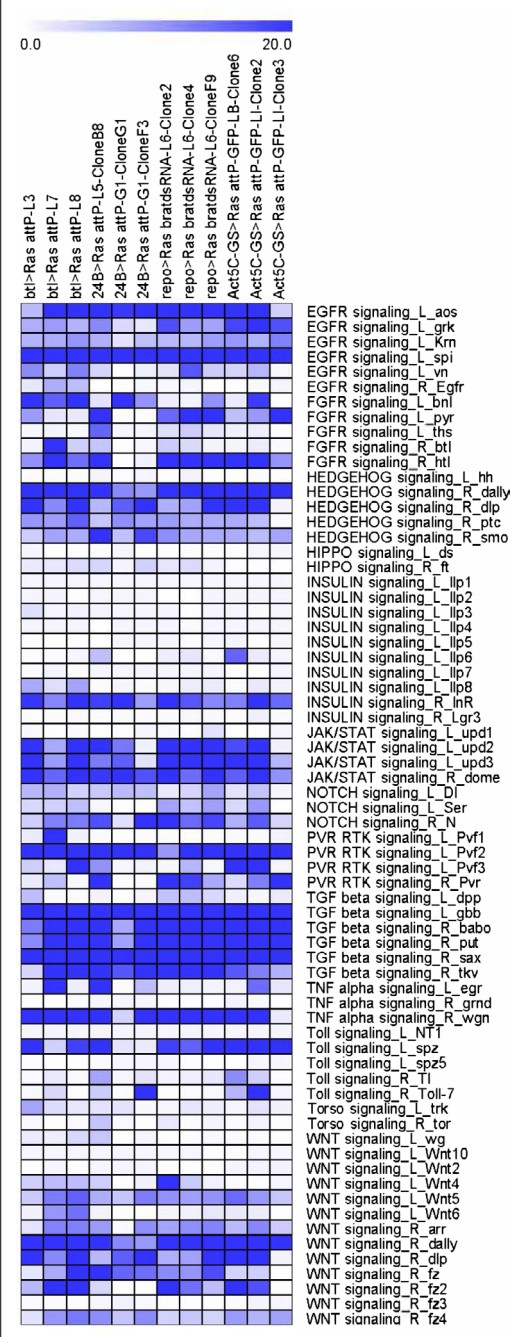

**Figure 2.** Expression levels of ligands and receptors for major signaling pathways. The ligand and receptor annotation for major signaling pathways was obtained from FlyPhoneDB (https://www.flyrnai.org/tools/fly_phone/web/). The expression levels of ligands and receptors are represented as a heatmap of FPKM values.

The online version of this article includes the following figure supplement(s) for figure 2:

**Figure supplement 1.** Comparison of lineage-restricted Ras cell lines with previously isolated *Drosophila* cell lines.

*Figure 2 continued on next page*

*Figure 2 continued*

**Figure supplement 2.** Principal component analysis (PCA) of RNAseq data from the lineage-restricted Ras cell lines.

**Figure supplement 3.** Relative expression of transcription factors associated in the literature with specific tissue lineages in the lineage-restricted Ras cell lines.

the squamous epithelial cells (Btl3 and Btl7) with ecdysone caused aggregation and formation of large multicellular clusters (*Figure 4*, *Figure 4—figure supplement 2*).

RNAseq data analysis comparing the top upregulated genes in the Btl cell lines with scRNAseq datasets revealed that the lines closely match the signatures of the adult trachea, a network of epithelial tubules (*Table 2*) and Btl3 expresses *tracheless* (*trh*) a master regulator of tracheal identity (*Wilk et al., 1996*; *Figure 2—figure supplement 3*). Overall, the morphological and molecular characteristics of the lines are consistent with an epithelial cell type of tracheal origin.

## Mesodermal-lineage cell lines

The *24B-Gal4* driver is an insertion in *held out wings* (*how*) and is expressed in mesoderm and muscle cells (*Brand and Perrimon, 1993*; *Zaffran et al., 1997*). Expression of *UAS-Ras^{V12}* with *24B-Gal4* readily produced continuous lines (*Table 1*, *Supplementary file 1*). Four clones (24B5-B8, 24B5-D8, 24BG1-F3, and 24BG1-G1) derived from two parental lines (24B5 and 25BG1) were analyzed in more detail (*Table 1*). The cells had a bipolar shape and expressed mesoderm markers including Twist and Mef2 (*Figures 1 and 5*, and *Figure 5—figure supplement 1*). When treated with ecdysone, cells from both parental lines and clones 24B5-B8 and 24B5-D8 elongated, fused as indicated by multinucleate cells, formed a network, and expressed Myosin heavy chain (Mhc) (*Figure 5* and *Figure 5—figure supplement 2*). There was also extensive cell lysis. Beginning 2 days after the second ecdysone treatment, the cells began to contract spontaneously. Contraction of cells from the 24B5 parental line and the two derivative clones (24B5-B8 and 24B5-D8) was visible in real time (*Videos 1 and 2*), whereas contraction of parental line 24BG1 cells was much slower and visualized more clearly in time-lapse (*Videos 3 and 4*). The clones 24BG1-F3 and 24BG1-G1 underwent morphological change but did not express Mhc or contract (*Figure 5—figure supplements 2 and 3*). In later passages,

**Table 2.** RNAseq data analysis.

| Tissue type | Cell line | Cell cluster scRNAseq | Enrichment p value scRNAseq | scRNAseq dataset | Cell type based on in situ data | Enrichment p value in situ |
|---|---|---|---|---|---|---|
| Glial | Rbr6-2 | Adult reticular neuropil-associated glial cell | 8.13E−05 | Whole body | Glia | 4.84E−05 |
| | Rbr6-4 | Cell body glial cell | 7.56E−04 | Whole body | | |
| | Rbr6-F9 | Adult glial cell | 8.13E−05 | Whole body | Glia | 3.42E−02 |
| Epithelial | Btl3 | Adult tracheal cell | 2.61E−06 | Whole body | Tracheal | 1.08E−01 |
| | Btl7 | Adult tracheal cell | 8.81E−04 | Oenocyte | | |
| | Btl8 | Adult tracheal cell | 2.72E−02 | Body | Tracheal | 2.05E−02 |
| Muscle | 24B5-B8 | Muscle cell | 2.93E−6 | Male reprod glands | | |
| | 24BG1-F3 | Muscle cell | 1.66E−04 | Antenna | | |
| | 24BG1-G1 | | | | Muscle | 8.83E−02 |
| Neuronal | ActGSI-2 | leg muscle motor neuron system | 5.79E−03 | Whole body | Neuron | 6.68E−02 |
| | ActGSB-6 | adult ventral nervous | 7.56E−04 | Whole body | Neuron | 5.71E−02 |
| Blood | ActGSI-3 | hemocyte | 1.00E−25 | Whole body | Circulatory system | 1.29E−01 |

Analysis using the *Drosophila* RNAi Screening Center's single-cell DataBase (DRscDB), all datasets used are from FCA 10x Sequencing (https://flycellatlas.org/). The in situ data were from the BDGP (https://insitu.fruitfly.org/cgi-bin/ex/insitu.pl) and the enrichment p value was calculated by a hypergeometric test.

the 24BG1 parental line also lost expression of Mhc and the ability to contract (*Figure 5—figure supplement 2*). This highlights the importance of using early passage cells and avoiding extended passaging that could alter the phenotypic (and genotypic) characteristics of the cells.

We also attempted to derive lines from Mef2-Gal4 because Mef2 regulates muscle development and is expressed in muscle progenitors and differentiated muscle suggesting *Mef2-Gal4* would be a good candidate for deriving cell lines (*Bour et al., 1995*; *Gossett et al., 1989*; *Lilly et al., 1995*; *Ranganayakulu et al., 1995*). However, only rare primary cultures had some proliferating cell patches, and none progressed to continuous lines (*Supplementary file 1*; *Figure 5—figure supplement 4*). Analysis of larvae from the cross (*Mef2-Gal4/+; UAS-GFP/UAS-Ras^V12^*) and control larvae (*Mef2-Gal4/+; UAS-GFP/+*) showed that Ras^V12^ expression disrupted muscle development, suggesting that the prevalent amorphous GFP-positive cells observed in primary cultures were abnormal muscle cells (*Figure 5—figure supplement 4*).

The RNAseq analysis for *24B-Gal4*-derived cell lines, identified the cells as muscle (*Table 2*). 24B5-B8 cells express high levels of the transcription factors *nautilus* (*nau*) and *twist* (*twi*) (*Figure 2—figure supplement 3*; *Figure 5—figure supplement 1*; *Table 2*), and high levels of *myoblast city* (*mbo*), which encodes an unconventional bipartite GEF with a role in myoblast fusion (*Erickson et al., 1997*). The capacity of these mesoderm-derived cell lines to differentiate into active muscle shows that the cells are muscle precursors and thus should be a useful reagent to analyze muscle physiology and development.

## Neuronal-like cell lines

To target neuronal cells, we expressed *UAS-Ras^V12^* with the pan-neural drivers *scratch-Gal4* and *elav-Gal4*, however none of the primary cultures resulted in continuous cell lines (*Supplementary file 1*; *Figure 6—figure supplement 1*). In previous work, we made primary cultures from embryos with ubiquitous expression of *UAS-Ras^V12^* using the *Act5C-Gal4* driver (*Simcox et al., 2008b*). The cells growing in these cultures included neuronal cells (*Simcox et al., 2008b*). Here, we used an *Act5C-GeneSwitch-Gal4* driver to express *UAS-Ras^V12^*. GeneSwitch-Gal4 is only active in the presence of the drug, RU486/mifepristone, which provides the advantage of being able to regulate Ras^V12^ expression

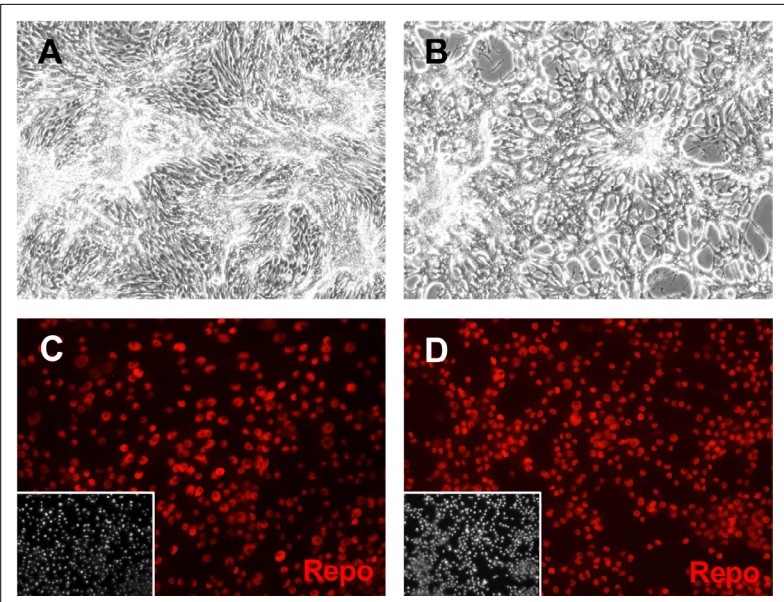

**Figure 3.** Glial clone Rbr6-2 cells express Repo. Cells were grown in plain medium (**A, C**) or treated with ecdysone (**B, D**). (**A, B**) After ecdysone treatment, cells make a lace-like network. (**C, D**) Cells express Repo with or without ecdysone treatment. Inset: DAPI (4',6-diamidino-2-phenylindole), DNA.

The online version of this article includes the following figure supplement(s) for figure 3:

**Figure supplement 1.** Marker gene expression in glial-lineage clones.

**Figure supplement 2.** Glial cell morphology with and without ecdysone treatment.

**Figure supplement 3.** Gross karyotypes of glial cell clones.

(*Nicholson et al., 2008*; *Osterwalder et al., 2001*). Several continuous lines were generated (*Supplementary file 1*). Clones derived from two of these (ActGSB-6 and ActGSI-2) (*Table 1*) were positive for the neuronal marker, HRP (horseradish peroxidase) (*Figure 6*, *Figure 6—figure supplements 2 and 3*). After differentiation with ecdysone, expression of Futsch/MAPB1 (*Hummel et al., 2000*) and Fas2 (*Mao and Freeman, 2009*) was enhanced and revealed axonal-like outgrowths from the cells (*Figure 6* and *Figure 6—figure supplement 3*). Differentiated cells also showed enhanced expression of Elav, which is commonly used as a marker for postmitotic neurons (*Figure 6* and *Figure 6—figure supplement 3*; *Robinow and White, 1991*). Elav is also expressed transiently in glial cells and proliferating neuroblasts *Berger et al., 2007*; however, the cells were negative for the glial marker Repo (*Supplementary file 2*).

RNAseq analysis revealed that many neuronal genes are upregulated in these cell lines, including *Glutamic acid decarboxylase 1* (*Gad1*), *slowpoke* (*slo*), *5-hydroxytryptamine (serotonin) receptor 1A* (*5-HT1A*), *Protein C kinase 53E* (*Pkc53E*), *Diuretic hormone 31 Receptor* (*Dh31-R*), and *straightjacket* (*stj*). In addition, comparison of the top upregulated genes in these cells to marker genes from scRNAseq data identifies a cell type of neuronal origin as the best match (*Table 2*). The cells should be a useful source of neuronal cells.

## Hemocyte-like cell line

Cells of clone ActGSI-3 derived from the ActGSI parental line (*UAS-Ras^{V12}* expression with *Act5C-GeneSwitch-Gal4*; *Table 1*, *Supplementary file 1*) show characteristics of hemocytes and express the hemocyte marker Hemese (*Figure 7*; *Kurucz et al., 2003*). They are also positive for HRP, but not other neuronal markers (*Figure 7—figure supplement 1*). ActGSI-3 cells divide in floating clusters, contrasting with S2 cells, which are also thought to be hemocytes, that grow as single cells (*Figures 1 and 7*).

RNAseq analysis demonstrated that many hemocyte genes are upregulated in these cells, including *serpent* (*srp*), *Hemese* (*He*), *eater*, *u-shaped* (*ush*), *Cecropin A2* (*CecA2*), and *Cecropin C* (*CecC*).

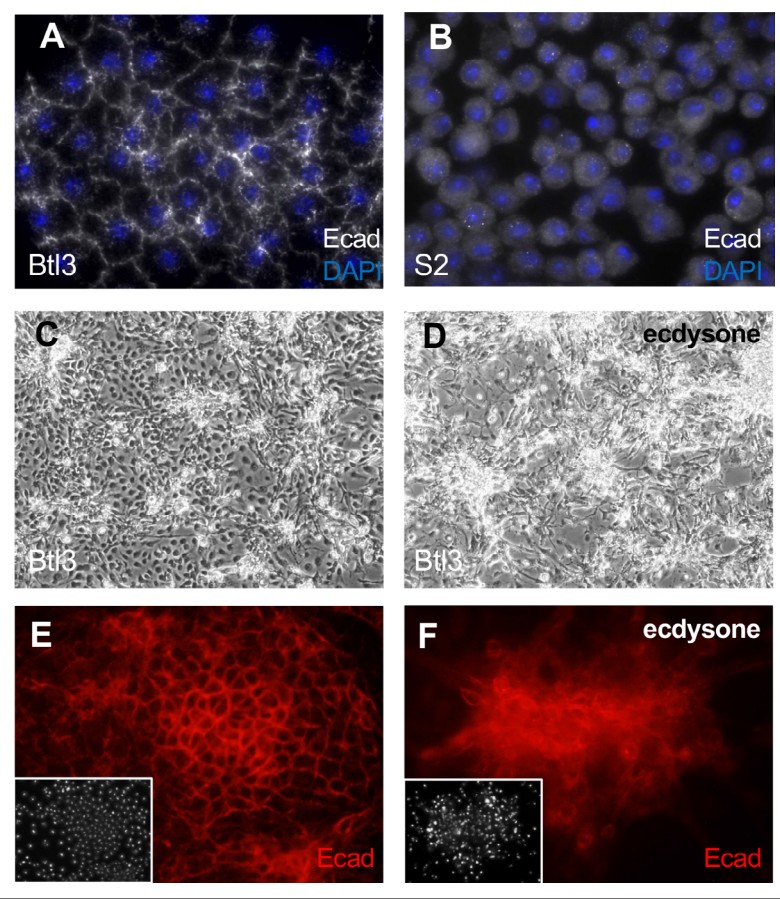

**Figure 4.** Tracheal-lineage cells of line Btl3 express the epithelial cadherin Ecad/Shotgun. All panels show Btl3 cells except (**B**) that shows S2 cells. Cells were grown in plain medium (**A–C, E**) or treated with ecdysone (**D, F**). (**A**) Btl3 cells form a squamous epithelial sheet and express Ecad/Shotgun at cell peripheries. (**B**) S2 cells grow as single cells and Ecad expression is diffuse. (**C**) Btl3 cells form a sheet with small cell clusters and expressed Ecad at the cell boundaries (**E**). (**D**) Ecdysone-treated cells form large multicellular clusters that expressed Ecad (**F**). Insets in E and F show nuclei with DAPI.

The online version of this article includes the following figure supplement(s) for figure 4:

**Figure supplement 1.** Marker gene expression in tracheal-lineage lines.

**Figure supplement 2.** Morphology of tracheal epithelial parental lines after ecdysone treatment.

**Figure supplement 3.** Gross karyotypes of tracheal epithelial parental cell lines.

Comparison of top upregulated genes with scRNAseq data showed that the cells have a strong match to the top marker genes of hemocytes (*Table 2*).

## Growth, karyotype, and transfection efficiency of cell lines

We determined the cell density at confluence for the cell lines (*Table 3*). The cells in each line grow to confluence attached to the tissue-culture surface, except ActGSI-3, which grow as floating cell clusters (*Figure 8*). The cells are not contact inhibited and cell clusters are formed allowing cells to grow to higher density (*Figure 8*). We determined the doubling time of 13 cell lines and clones using growth curves (*Table 3*; *Figure 8—figure supplement 1*). Most had doubling times within a range of approximately 20–40 hr (*Table 3*). The hemocyte-like clone ActGSI-3 was an outlier with a longer doubling time of 70 hr (*Table 3*). In cells from clones ActGSB-6, ActGSI-2, and ActGSI-3, expression of Ras$^{V12}$ is dependent on GeneSwitch Gal4, which is active only in the presence of mifepristone. In the absence of the drug the cells become quiescent (*Figure 8—figure supplement 1*).

We determined the gross karyotype of 13 cell lines and clones. In keeping with previous findings for Ras$^{V12}$ expressing cell lines, most (8) were diploid, or near diploid (*Simcox et al., 2008b*; *Table 3*;

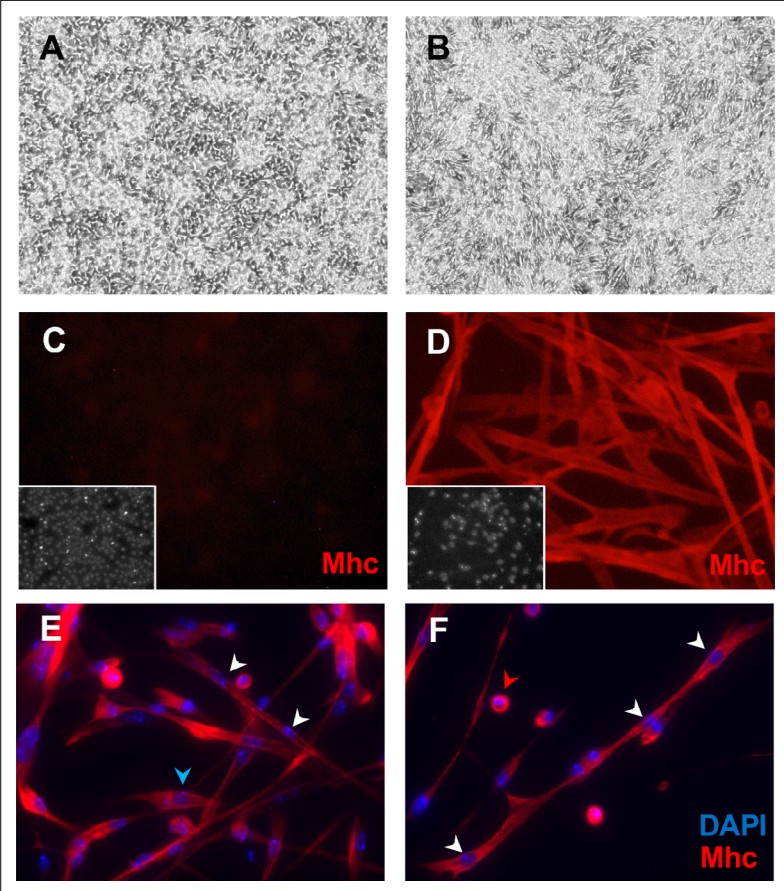

**Figure 5.** Mesodermal-lineage cells of Clone 24B5-B8 express Myosin heavy chain after differentiation. Cells were grown in plain medium (**A, C**) or treated with ecdysone (**B, D–F**). (**A**) Cells have a bipolar shape. (**B**) Ecdysone-treated cells elongate and contract. (**C**) Control cells do not express Mhc. (**D**) Ecdysone-treated cells express the muscle marker Mhc. Inset: DAPI, DNA. (**E, F**) Differentiated 24B5-B8 fuse to form muscle fibers that contain multiple nuclei (white arrowheads), some differentiate without fusing with other cells and have single nucleus (blue arrowhead), and some fail to differentiate and remain spherical with a single nucleus (red arrowhead).

The online version of this article includes the following figure supplement(s) for figure 5:

**Figure supplement 1.** Marker gene expression in mesodermal-lineage clones.

**Figure supplement 2.** Immunostaining of mesodermal-lineage cells for Myosin heavy chain.

**Figure supplement 3.** Mesodermal cells showed altered morphology after ecdysone treatment.

**Figure supplement 4.** Mef2-Gal4; UAS-GFP; UAS-Ras$^{V12}$ cultures.

**Figure supplement 5.** Gross karyotypes of mesodermal cell clones.

---

*Figure 3—figure supplement 3*; *Figure 4—figure supplement 3*; *Figure 5—figure supplement 5*; *Figure 6—figure supplement 4*; *Figure 7—figure supplement 2*). Related clones had similar karyotypes, which likely indicates that parental lines may also be clonal as a result of selective pressure for cells that grow well in culture. Some lines were polyploid and common aneuploid conditions include loss of an X chromosome and varying numbers of chromosome 4 (*Table 3*).

Nine parental and clonal lines were transfected with an Act5C-EGFP plasmid and the fraction of GFP-positive cells was determined after 48 hr. Cells from all lines tested could be transfected. The range of efficiency was from 16% to 34% with most lines showing transfection of approximately one quarter of the cells (*Table 3*). Similarly treated, cells from the S2 line showed an efficiency of 53%.

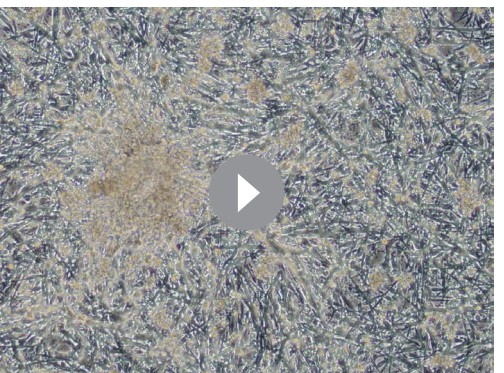

**Video 1.** 24B-Gal4B5-B8 cells contract spontaneously after differentiation with ecdysone.
https://elifesciences.org/articles/85814/figures#video1

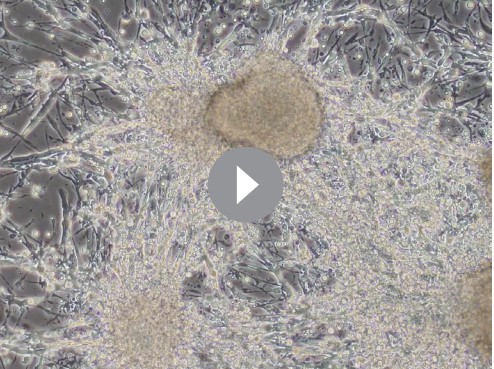

**Video 3.** 24B-Gal4GI cells contract spontaneously after differentiation with ecdysone. Time-lapse video, view looping.
https://elifesciences.org/articles/85814/figures#video3

## Discussion

Expressing activated Ras, Ras$^{V12}$, in primary cells provides a growth and survival advantage and leads to the rapid and reliable generation of continuous cell lines—the so-called Ras method (*Simcox et al., 2008b*). In a second-generation version of the Ras method, we found that restricting Ras$^{V12}$ expression with lineage-specific Gal4 drivers gave the targeted cells a competitive advantage and produced continuous lines with expected cell-type-specific phenotypes. With this approach we produced glial, epithelial, and muscle cell lines using the *repo-*, *btl-*, and *24B/how-Gal4* drivers, respectively.

In theory, the approach could be used to produce cell lines corresponding to any cell type for which there is an appropriate Gal4 driver. We tried to derive lines with *Mef2-Gal4*, a muscle master regulator gene, and the pan-neuronal driver *elav-Gal4*; however, no continuous lines were produced (*Supplementary file 1*; *Figure 5—figure supplement 4* and *Figure 6—figure supplement 1*). In both cases, Ras$^{V12}$ expression appeared to disrupt growth of the targeted cell type. In the case of the muscle lineage, *24B/how-Gal4* was efficient at producing cell lines. The success with one and not the other muscle driver shows that in practice, it may be necessary to test multiple Gal4 lines for a given lineage. Drivers with very specific expression patterns may prove useful, including those generated by the Split Gal4 system (*Luan et al., 2006*). As with any tissue-culture system, the unnatural conditions of growing in vitro may select for 'generic' cells that survive well in culture and lose their lineage identities. This means that characterizing cell lines after generation for a battery of features (morphological, physiological, and molecular) is an essential step in assessing whether cells represent the tissue of origin expected for a given Gal4 driver.

*repo-Gal4* is a pan-glial driver and many primary cultures expressing Ras$^{V12}$ with this driver reached confluence and could be passaged several times

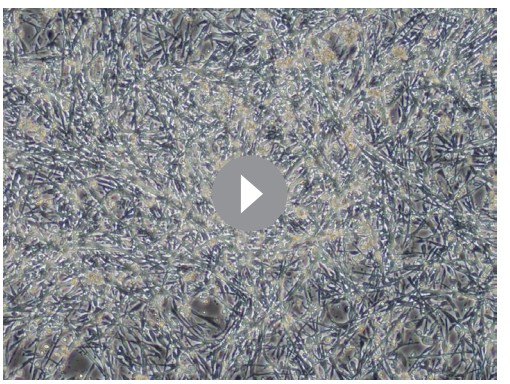

**Video 2.** 24B-Gal4B5-B8 cells contract spontaneously after differentiation with ecdysone.
https://elifesciences.org/articles/85814/figures#video2

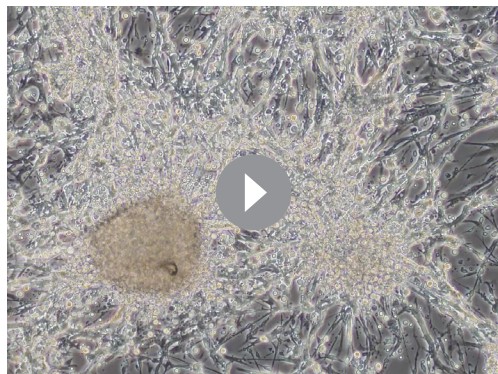

**Video 4.** 24B-Gal4GI cells contract spontaneously after differentiation with ecdysone. Time-lapse video, view looping.
https://elifesciences.org/articles/85814/figures#video4

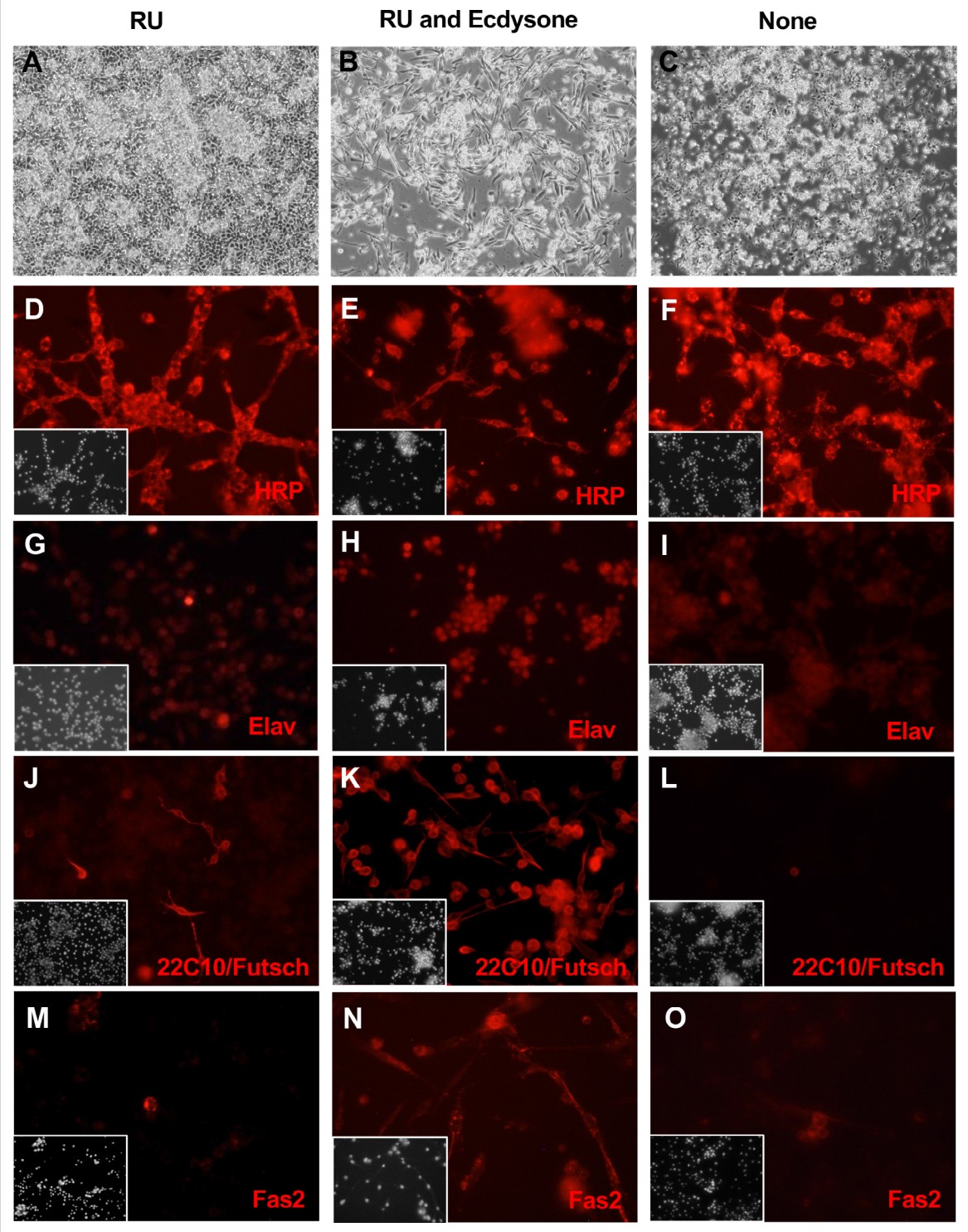

**Figure 6.** Neuronal-like clone ActGSI-2 expresses neuronal markers. ActGSI-2 cells were grown in three conditions: RU486 (**A, D, G, J, M**); RU486 and ecdysone (**B, E, H, K, N**), or with no additives (**C, F, I, L, O**). RU486/mifepristone is required for GeneSwtch-Gal4 activation, transgene expression, and cell proliferation. (**A**) In the growing condition, cells reach confluence and continue to grow by piling up. (**B**) After ecdysone treatment cells elongated and developed axonal-like outgrowths. (**C**) In the quiescent state (no RU), cells do not proliferate and fail to reach confluence. (**D–F**) Cells in all conditions are positive for HRP. (**G–I**) Expression of Elav, is elevated after ecdysone treatment (**H**). (**J–L**) Expression of Futsch/MAP1B-like protein (recognized by antibody 22C10) is elevated after ecdysone treatment (**K**). (**M–O**) Fas2 neural-adhesion protein. Cells show elevated expression after ecdysone treatment (**N**). Insets: DAPI, DNA.

*Figure 6 continued on next page*

*Figure 6 continued*

The online version of this article includes the following figure supplement(s) for figure 6:

**Figure supplement 1.** elav-G4; UAS-GFP; UAS-Ras$^{V12}$ cultures.

**Figure supplement 2.** Marker gene expression in neuronal-like clones.

**Figure supplement 3.** Neuronal-like clone ActGSB-6.

**Figure supplement 4.** Gross karyotypes of neuronal-like cell clones.

but did not produce continuous lines (***Supplementary file 1***). We tested different genotypes to determine if the success rate could be improved by modulation of Ras$^{V12}$ expression (co-expression of the Gal4 inhibitor Gal80ts), co-expression of the p35 baculovirus survival factor, or growth stimulation by downregulation of tumor suppressors (dsRNA for *warts* or *brat*). One line, also harboring a *Gal80ts* transgene, reached passage 25; however, the line was unstable and in early passages the cells variably lost Repo expression and changed morphologically. The one continuous glial line generated expresses a transgene that targets the tumor suppressor, *brat* (repo-Gal4; UAS-Ras$^{V12}$; UAS-brat$^{dsRNA}$). Given a single success, it is not clear if downregulation of *brat* contributed to derivation of the line. Moreover, there is no evidence that these genotypic variations enhanced cell line generation with other drivers, as primary cultures expressing Ras$^{V12}$ without modulation or a survival factor produced lines with similar success rates for the *btl-Gal4* or *24B/how-Gal4* drivers (***Supplementary file 1***).

As with all types of tissue culture, best practices involve maintaining frozen aliquots of cell lines at relatively low passage numbers. Aliquots of cells from the lines and clones described here, on which RNAseq was performed, have been archived at similar passage numbers as those used for the RNAseq analysis. This will allow users to start experimentation with the lines in a known state. The importance of this is exemplified by line 24BG1, which lost the ability to contract and express the muscle protein Mhc after multiple passages (***Figure 5—figure supplement 2***).

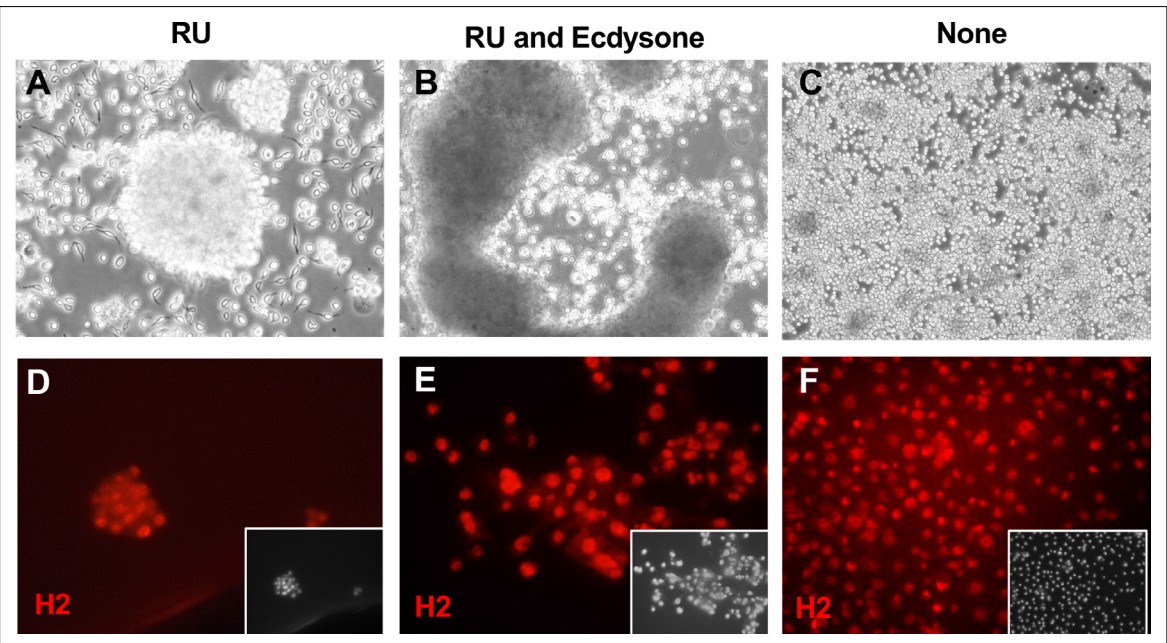

**Figure 7.** Hemocyte-like Clone ActGSI-3 morphology and marker expression. Cells were grown in three conditions: RU486 (**A, D**); RU486 and ecdysone (**B, E**), or with no additives (**C, F**). (**A**) In the growing condition, cells formed floating clusters of multiple cells. (**B**) After ecdysone treatment cells formed large aggregates and there was cell lysis. (**C**) In the quiescent state (no RU), individual round cells are seen. (**D–F**) Cells in all conditions express the hemocyte cell marker Hemese, as recognized by the antibody H2. Inset: DAPI, DNA.

The online version of this article includes the following figure supplement(s) for figure 7:

**Figure supplement 1.** Marker gene expression in hemocyte-like clone.

**Figure supplement 2.** Gross karyotypes of hemocyte cell clone.

**Table 3.** Confluent density, growth, karyotype, and transfection efficiency of cell lines.

| Tissue type | Line | Confluent density (×10⁶)* | Doubling time (hr) | Karyotype | Transfection efficiency (%) |
|---|---|---|---|---|---|
| Glial | Rbr6-2 | 1.8 | 20 | 8, XY | 24 |
| | Rbr6-4 | 2.4 | 20 | 8, XY | 28 |
| | Rbr6-F9 | 3.4 | 19 | 8, XY | 22 |
| Epithelial | Btl3 | 3.7 | 33 | 7, XY, –4 | 26 |
| | Btl7 | 2.6 | 37 | Abnormal tetraploid, XX, variable 4 | 34 |
| | Btl8 | 2.7 | 22 | Abnormal tetraploid, XX, –4 | 16 |
| Mesodermal | 24B5-B8 | 1.4 | 29 | Abnormal tetraploid, XXY, variable 4 | 23 |
| | 24B5-D8 | 5.1 | 23 | Abnormal tetraploid, XX, variable 4 | 27 |
| | 24BG1-G1 | 2.8 | 21 | 8, XY (some –4) | ND |
| | 24BG1-F3 | 2.7 | 35 | 8, XY (some –4) | ND |
| Neuronal | ActGSB-6 | 2.9 | 23 | 7, XO | 29 |
| | ActGSI-2 | 8.1 | 27 | 8, XX | ND |
| Blood | ActGSI-3 | 1.9 | 70 | Abnormal tetraploid, XX, variable 4 | ND |
| | S2 | 6.2 | ND | ND | 53 |

*Confluent density in one well of a 12-well plate, 3.5 cm² surface area (average of three wells).

The mesodermal, neuronal, and glial cells represent in vitro counterparts of the tissues of origin that can be used for studying development and physiology in an accessible and reproducible system. The mesodermal cells that differentiate into active muscle will allow investigation of muscle fusion, as the cells are multinucleate (*Figure 5*), as well as muscle physiology and function. For example, the cells contract spontaneously and in apparent waves (*Videos 1 and 2*); however, the mechanism for stimulation (if any) and regulation have not been investigated and may cast light on in vivo processes. Given a variety of cell types, it will also be interesting to examine cell form and function in co-cultures, for example, of glia and neurons.

The method and the cells will be useful for generating disease models. New lineage-specific lines could be generated in the desired mutant background by establishing primary cultures from embryos in which only the mutant genotype expresses Ras$^{V12}$ giving these cells a growth and survival advantage (*Simcox et al., 2008a*). Derivative lines should include those of the desired cell type and genotype. Alternatively, the existing cell lines could be edited using CRISPR, or insertion of transgenes using the attP site that most lines and clones contain (*Supplementary file 1*; *Bateman et al., 2006*; *Manivannan et al., 2015*).

The cells with epithelial morphology derived from the tracheal lineage (Btl3 and Btl7) will provide good models for investigating assay conditions that promote polarization and 3D cell interactions that could allow the cells to manifest a more complex tissue architecture. In keeping with this possibility, treating these cells with ecdysone to induce differentiation showed cell clumping suggestive of a multicellular structure (*Figure 4—figure supplement 2*).

RNAseq analysis of cells from the ActGSI-3 cell clone showed a striking similarity to hemocytes, and the cells may be a good model for studying immunity (*Table 2*). The cells lyse after ecdysone treatment suggesting they are of embryonic origin (*Figure 7*). The cells grow as floating cell clumps (*Figures 1 and 7*) that may recapitulate subepidermal clusters of sessile hemocytes of the larva (*Leitão and Sucena, 2015*; *Márkus et al., 2009*).

The most significantly upregulated marker genes in each cell line are significantly enriched for top marker genes from expected cell types based on the single-cell RNAseq data from Fly Cell Atlas in

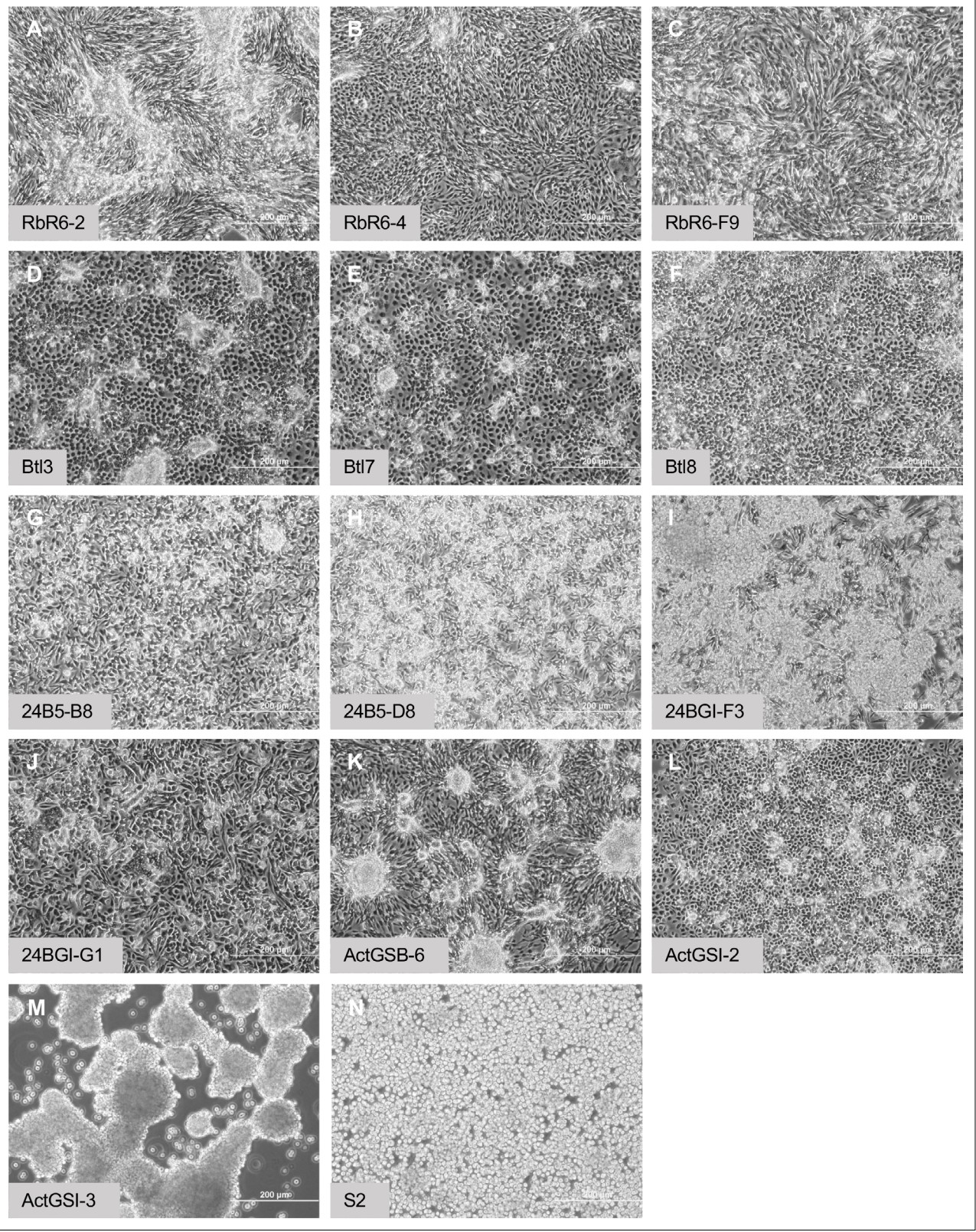

**Figure 8.** Morphology of confluent cultures. (**A–C**) Glial-lineage clones. The cells grow in dense sheets and ridges with swirl patterns. (**D–F**) Tracheal-lineage cells. Btl3 and Btl7 cells form squamous epithelial sheets with raised clusters of cells. Btl8 grow densely however individual cells remain separate. (**G–J**) Mesodermal-lineage cells. The cells grow densely, and form raised clusters. (**K, L**) Neuronal-like clones. ActGSB-6 cells grow densely and form

*Figure 8 continued on next page*

*Figure 8 continued*

peaks and valleys. ActGSI-2 cells grow densely with scattered raised clusters. (**M**) Hemocyte-like clone ActGSI-3. The cells form floating clusters that coalesce into large cell rafts. (**N**) Schneider's S2 cells. The cells grow to high density in suspension. Scale bar = 200 µm.

The online version of this article includes the following figure supplement(s) for figure 8:

**Figure supplement 1.** Growth curves.

most cases. This indicates the potential value of these cell lines as corresponding in vitro models for studying these cell types. While the cells will prove to be valuable models, it should be noted that even those showing a clear differentiated phenotype exhibit unexpected patterns of gene expression. For example, some cells in the mesodermal clone, 24B5-B8, are positive for HRP (*Figure 5—figure supplement 1*; *Supplementary file 2*) and the two neuronal-like lines express a mesodermal marker, Twist (*Figure 6—figure supplement 2*; *Supplementary file 2*). This anomalous gene expression is likely to be an effect of Ras activation on downstream pathways and genes. Ras/MAPK has a key role in muscle cell determination (*Buff et al., 1998*; *Carmena et al., 2002*; *Halfon et al., 2000*) and activates downstream muscle determination genes. It will also be important to consider what genes are not expressed by a given cell line, for example, *glial cell missing* (*gcm*) is not differentially expressed in the three glial-lineage cell clones and *gcm2* is differentially expressed in only two of the three clones. Further, *tracheless* (*trh*) is only differentially expressed in one of the three tracheal-lineage cell lines. Similarly, the muscle-specific transcription factors *twist* (*twi*), *nautilus* (*nau*), *snail* (*sna*), and *Mef2* show variable expression in the muscle-lineage cell clones. It should also be noted that the expression patterns were determined for undifferentiated cells and expression levels could change after hormone exposure.

The cells will have value for both low- and high-throughput approaches, including genetic or compound screens for which screening in the relevant cell type will result in identifying targets that are more likely to be of physiological relevance. Most of the cells have an attP-flanked cassette (*Table 1*), which makes them amenable to insertion of transgenes such as reporters by Recombination Mediated Cassette Exchange (RMCE) (*Bateman et al., 2006*; *Manivannan et al., 2015*). Moreover, cells competent for RMCE can be modified by stable expression of Cas9 and then used for genome-wide CRISPR pooled screening. With this approach, a library of single guide RNAs (sgRNAs) are integrated at RMCE sites (*Viswanatha et al., 2018*; *Viswanatha et al., 2019*). This generates a pool of cells, each with a different sgRNA, that can be subjected to a screen assay. Results are identified by PCR amplification of inserted sgRNAs followed by next-generation sequencing to detect sgRNAs that are enriched or depleted in the experimental cell pool as compared with a control. To date, pooled CRISPR screens in *Drosophila* have only been performed in S2 cells, which have hemocyte-like features. The availability of new cell lines with muscle, glial, and epithelial characteristics will enable screens designed to interrogate biological processes specific to these cell types.

There are hundreds of *Drosophila* cell lines; however, the number corresponding to known cell types is low. This is due in part to the lack of a method for generating cell lines from specific tissues. We expect that the method described here, using restricted expression of Ras$^{V12}$, will be a tractable approach for investigators to generate lines of cell types of interest. Single-cell cloning followed by cell characterization (immunohistochemistry and RNAseq) also proved to be a useful method to identify cell-type-specific lines and this approach could identify additional valuable lines in the existing collection at the DGRC. In summary, we show that lineage-restricted Ras expression and cell cloning has produced a set of new cell lines that will be of immediate value for analyses in the five cell types they represent.

## Materials and methods
### Fly stocks

The following fly stocks were used to create primary cell lines: *Gal4 drivers*: 24B/how-Gal4, w[*]; P(w[+mW.hs]=GawB)how[24B] (BL 1767); repo-Gal4, P(GAL4)repo (BL 7415); btl-Gal4, P(GAL4-btl.S)3-2 (BL 78328); Act5C-GeneSwitch-Gal4, P(UAS-GFP.S65T)Myo31DF[T2]; P(Act5C(-FRT)GAL4.Switch.PR)3 (BL 9431). *Transgenes*: UAS-Ras$^{V12}$ (3), P(w[+mC]=UAS-Ras85D.V12)TL1 (BL 64195); UAS-Ras$^{V12}$ (2), P(w[+mC]=UAS-Ras85D.V12)2 (BL 64196); UAS-Ras$^{V12}$ with RMCE site (3), P(w[+mC]=UAS-Ras85D.

V12)TL1, P(w[+mC]=attP.w[+].attP)JB89B (BL 64197); UAS-GFP nuclear, P(UAS-GFP.nls)14 (BL 4775); brat[dsRNA], P(y[+t7.7] v[+t1.8]=TRiP.HMS01121)attP2 (BL 34646); UAS-p35 baculovirus death inhibitor, P(w[+mC]=UAS-p35.H)BH1 (BL 5072) and Gal80ts, w[*]; P(w[+mC]=tubP-GAL80[ts])20 (BL 7019).

## Setting up primary cultures

This follows a detailed method, which has additional information (*Simcox, 2013*), except that no yeast paste is used on the egg collection plates. Yeast paste, even when sterilized, promotes contamination in the cultures. Crosses were made between the Gal-4 driver lines and UAS-Ras[V12] lines. Some Ras[V12] stocks had additional alleles as noted in *Supplementary file 1*. Approximately 200 males and 200 females of a cross were transferred into a laying cage, with a fluted Whatman 3MM paper insert to increase surface area, and eggs were collected using 60-mm Petri dishes containing egg laying medium. Egg collections were made during the day for 8 hr at room temperature or 16 hr overnight at 17°C. After collection, approximately 3 ml of TXN (NaCl [0.7%], Triton X [0.02%] in water) was added to the plate. Any hatched larvae, which rise to the surface, were removed and the unhatched embryos were dislodged using a large soft paint brush to gently release them from the surface. Embryos were tipped off with the liquid into a sieve. Additional rinsing and brushing were used to ensure most embryos were dislodged and collected in the sieve. After thorough rinsing of the embryos with TXN from a squirt bottle, the sieve was upended over a 15-ml Falcon tube and a stream of TXN was used to transfer the embryos into the tube. Once the embryos settled, the TXN was removed and replaced with 3 ml of 50% bleach (Clorox) in water. The tube was capped and inverted three to five times and subsequently the embryos were treated using sterile techniques. The embryos were allowed to settle at the bottom of the tube and the bleach was removed after 3–5 min. The bleach dechorionates and surface sterilizes the embryos. The embryos were rinsed 2× with 4 ml of sterile TXN and transferred to a fresh tube of TXN to minimize bleach contamination. After two additional TXN rinses the embryos were transferred to TXN in a 5-ml glass homogenizer (with Teflon pestle). Embryos were rinsed in 3 ml of water followed by a rinse in 1 ml of Schneider's S2 medium (supplemented with 10% heat inactivated fetal bovine serum and 1× Pen-strep solution). Embryos tend to clump in the Schneider's S2 medium and stick to the sides of the homogenizer and pipette and care is needed to remove the medium without disturbing the embryos. 3 ml of fresh Schneider's S2 medium was added to the homogenizer and the embryos were disrupted by three gentle strokes with the pestle. Care was taken to minimize bubbles by not withdrawing the pestle beyond the surface of the liquid. The homogenate was allowed to settle for 2 min and the supernatant was transferred to a 15-ml Falcon tube leaving the large cell clumps and any whole embryos in the bottom of the homogenizer. 3 ml of fresh Schneider's S2 medium was added to the homogenizer and three more strokes, with a twist at the bottom, were used to disrupt remaining tissue and embryos. The second homogenate was added to the Falcon tube. The tube was centrifuged in a benchtop centrifuge at $1400 \times g$. The supernatant was discarded, and the pellet was resuspended in 3-ml Schneider's S2 medium and centrifugation step and washing with Schneider's S2 medium was repeated twice more. The final pellet size was estimated and plated in 1 or more 12.5 cm$^2$ T-flasks with 2–3 ml Schneider's S2 medium. The number of flasks needed for a given pellet size can also be estimated from the volume of packed embryos with approximately 30 µl of packed embryos being sufficient for one flask.

## Culture conditions for new cell lines

Cells were grown in 25 cm$^2$ T-flasks at 25°C in Schneider's S2 medium and were passaged at between 90% and full confluence (*Figure 8*) using trypsin to release cells from the tissue-culture surface. Trypsin is needed as cells in all the lines are adherent except ActGSI3 cells that float freely (*Figures 1 and 8*). Cells were pelleted and approximately 20–25% of the cells were plated in a new flask. Cells were checked using an inverted microscope approximately every 5 days. The medium was changed on cultures showing signs of poor cell health (extended processes, little growth). This was sometimes necessary for cell types that are more metabolically active and acidify the medium, including the mesodermal lines. Cells were passaged every 5–7 days. Cell freezing (Schneider's S2 medium with 20% heat inactivated fetal bovine serum and 10% DMSO (Dimethyl sulfoxide)) was used to keep a supply of frozen aliquots so that cells with similar passage numbers were used in experiments.

## Cell cloning

For puromycin selection, 2–6 × 10⁵ cells in a 35-mM well were transfected with 0.4 µg of DNA encoding a puro resistance plasmid (pCoPURO, Addgene #17533) using Effectene Transfection Reagent (QIAGEN). After 24 hr, cells were selected with puromycin at 0.5–2.5 µg/ml for 5 days. After 2–4 weeks, colonies were isolated and expanded. For dilution cloning, cells were seeded into a 96-well plate at a concentration of 0.5–1 cell/well in 100 µl conditioned media (*Housden et al., 2015*).

## Hormone treatment

To simulate the major pulse of ecdysone at the larval to pupal transition, cells were treated with two 24 hr doses of β-ecdysone (Sigma 5289-74-7) at 1 µg/ml separated by 24 hr in non-supplemented medium.

## Immunohistochemistry

Cells were fixed with 4% paraformaldehyde (Electron Microscopy Sciences) for 15 min or 3.5% formaldehyde (Sigma) for 30 min at room temperature, and then rinsed twice with 0.1% Tween-20 in phosphate-buffered saline (PBS-T). Cells were permeabilized (0.2% Triton X-100 in PBS) for 10 min at room temperature. Cells were blocked (5% bovine serum albumin in PBS-T) for 30 min at room temperature and incubated with diluted primary antibodies overnight at 4°C. Cells were washed three times with PBS-T and incubated with diluted secondary antibodies in blocking buffer for 1 hr at room temperature or overnight at 4°C. Cells were washed three times with PBS-T and mounted in Vecta-Shield with DAPI (Vector Laboratories). For the Dcad2 antibody, cells were fixed and processed as described in *Oda et al., 1994*. The following primary antibodies and dilutions were used: HRP (rabbit polyclonal, Jackson ImmunoResearch 323-005-021, 1:500), 22C10 (mouse monoclonal anti-Futsch, Developmental Studies Hybridoma Bank, DSHB, 1:100), ELAV (rat monoclonal, DSHB 7E8A10, 1:100), Repo (mouse monoclonal, DSHB 8D12, 1:100), FasII (mouse monoclonal, DSHB 1D4, 1:100), Twist (a gift from M. Levine, UC Berkeley, CA, guinea pig 1:500), MHC (mouse monoclonal, DSHB 3E8-3D3, 1:100), Dcad2 (rat monoclonal, DSHB, 1:100), and DMef2 (a gift from J. R. Jacobs [*Vanderploeg et al., 2012*], rabbit polyclonal, 1:500), H2 (mouse monoclonal, [*Kurucz et al., 2003*], 1:10). Cells were incubated with the following secondary antibodies at the indicated dilutions: Cy3-conjugated goat anti-mouse (Jackson ImmunoResearch 115-165-003, 1:1000), Cy3-conjugated goat anti-rat (Jackson ImmunoResearch 112-165-003, 1:1000), Cy3-conjugated goat anti-guinea pig (Jackson ImmunoResearch 106-165-003, 1:1000), Cy3-conjugated goat anti-rabbit (Jackson ImmunoResearch 111-165-045, 1:1000), and Alexa Fluor 488-conjugated donkey anti-rabbit (Invitrogen A-21206, 1:1000).

## Growth curve analysis

1–2 × 10⁵ cells were plated in a 12-well plate. Cells were counted from triplicate wells every 3 days over a 9-day period. Doubling time was calculated using log2 cell numbers (*Roth, 2006*).

## Karyotype analysis

Cells were grown to 50–90% confluence and incubated with 0.05 µg/ml KaryoMAX (Gibco-Thermo Fisher 15212012) for 3–18 hr. Cells were processed for analysis using the method in *Lee et al., 2014*, which uses 0.5% sodium citrate as a hypotonic solution and a 3:1 ice cold mix of methanol and acetic acid as a fix. After dropping fixed cells, slides were air dried and mounted in VectaShield with DAPI (Vector Laboratories) and viewed with an Olympus BX41 microscope.

## Transfection

Cells in a 6-well plate (approximately 70% confluent) were transfected with 0.4 µg of an Actin5C-EGFP plasmid (pAc5.1B-EGFP, Addgene #21181) using Effectene Transfection Reagent (QIAGEN). The fraction of GFP-positive cells was scored after 48 hr.

## RNA extraction and RNAseq

Cell cultures were grown and expanded in their respective media. All cell lines were cultured in Schneiders *Drosophila* Medium (Gibco Cat # 21720001), supplemented with 10% fetal bovine serum (Cytiva Hyclone Cat SH30070.03). For Act5C-GS>Ras attP-GFP-LI-Clone 2, Act5C-GS>Ras attP-GFP-LI-Clone 3, and Act5C-GS>Ras attP-GFP-LB-Clone 6 cultures were grown in the same basal media

supplemented with 10 nM of Mifepristone (Thermo Fisher Cat# H11001). Cultures were allowed to grow in T-25 flasks to become confluent before treatment with trypsin (Gibco Cat# 12604013) for 4 min to dislodge the cell monolayer from the growth surface. The cells were resuspended in 4 ml of their respective media and 1 ml of the cell suspension was collected for pelleting, followed by washing in 1× PBS, and then flash-freezing in liquid nitrogen. All cell samples were processed in triplicates.

Total RNA was isolated from the pellets using the TRIzol reagent (Life Technologies [Ambion], Cat#:15596018) as per the manufacturer's instructions. The isolated total RNA was subjected to further purification using the RNeasy Mini Kit (QIAGEN, Cat#74104) and the RNA post-cleanup was eluted in RNase-free water. The eluted total RNA was confirmed to have a $A_{260}/A_{280}$ ratio >1.8 and RIN >7.

Upon passing the quality control parameters, Illumina TruSeq libraries were constructed using TruSeq stranded mRNA HT kit (Illumina, Cat# RS-122-2103). Paired end sequencing was performed on an Illumina NextSeq 500 with a 150-cycle high output kits (Illumina, Cat# FC-404-2002).

### RNAseq data analysis

Raw data processing was performed using the STAR sequence aligner (https://github.com/alexdobin/STAR; *Dobin et al., 2013*). Reads were aligned to the *Drosophila* genome and featureCounts were used to get gene counts from all samples into a count matrix for downstream analysis. A principal component analysis plot was produced using heatmaply. FPKM values were calculated using fpkm(-DEseq2) using gene length output by featureCounts. The reference genome used was FB2022_05, dmel_r6.48 (FlyBase) (*Jenkins et al., 2022*). Both raw sequencing reads and the count matrix were deposited in the NCBI Gene Expression Omnibus (GEO) database under the accession number GSE219105. The processed dataset has also been imported into DGET database for user to mine gene(s) of interest or search for genes with similar expression pattern (https://www.flyrnai.org/tools/dget/web/).

Each sample was compared against all other samples by using DESeq2 ( *Love et al., 2014*) to determine differentially expressed genes (DE calling). The set of top DE genes for each cell line was compared with the top 100 markers in single-cell RNAseq datasets corresponding to cell types in the Fly Cell Atlas 10× datasets (*Li et al., 2022*). Enrichment analysis was conducted using the DRscDB tool to identify the Fly Cell Atlas cell type that matched closely to each cell line (*Hu et al., 2021*). We also compared the DE genes with the genes identify in various tissues in embryo and larval based on in situ data (PMID: 24359758, 17645804, 12537577) and majority of the best matching tissues are consistent with the analysis using scRNAseq datasets (*Table 2*).

The RNAseq data for the cell lines described in this work were also compared with RNAseq datasets determined previously for 24 other *Drosophila* cell lines (*Cherbas et al., 2011*). The comparison was conducted by hierarchical clustering analysis using Pearson correlation coefficient scores. To survey the activities of major signaling pathways in the cell lines, we specifically selected the ligands and receptors annotated at FlyPhoneDB (PMID: 35100387) to plot their expression levels using heatmap.

### Materials availability

All cell lines described here have been deposited to the *Drosophila* Genomics Resource Center (DGRC) at Indiana University. The lines are available for distribution to the research community.

## Acknowledgements

We thank M Levine, J R Jacobs, and D Hultmark for antibodies and the Bloomington Stock Center for fly stocks. We thank Mikhail Kouzminov for help with data analysis. Funding This work is supported by the National Institutes of Health (NIH Office of the Director R24 OD019847 to NP, SEM, and AS, P40OD010949 to the DGRC, and NIH NIGMS P41 GM132087 to the DRSC-BTRR), the National Science Foundation (IOS 1419535 to AS, and support while serving at the National Science Foundation to AS), the Howard Hughes Medical Institute (NP), and a grant from Women & Philanthropy at the Ohio State University (to AS).

## Additional information

### Funding

| Funder | Grant reference number | Author |
| --- | --- | --- |
| National Institutes of Health (NIH) Office of the Director | R24 OD019847 | Norbert Perrimon Stephanie E Mohr Amanda Simcox |
| National Institutes of Health | P40OD010949 | Andrew Zelhof |
| National Institutes of Health | P41 GM132087 | Norbert Perrimon Stephanie E Mohr |
| National Science Foundation | IOS 1419535 | Amanda Simcox |
| Howard Hughes Medical Institute | | Norbert Perrimon |
| Women & Philanthropy at The Ohio State University | Grant | Amanda Simcox |
| National Science Foundation | Support while serving at the National Science Foundation | Amanda Simcox |

The funders had no role in study design, data collection, and interpretation, or the decision to submit the work for publication. Any opinion, findings, and conclusions or recommendations expressed in this material are those of the authors and do not necessarily reflect the views of the National Science Foundation.

### Author contributions

Nikki Coleman-Gosser, Shane Stitzinger, Formal analysis, Investigation, Writing – review and editing; Yanhui Hu, Formal analysis, Writing – review and editing; Shiva Raghuvanshi, Molly Josifov, Investigation; Weihang Chen, Formal analysis; Arthur Luhur, Daniel Mariyappa, Investigation, Writing – review and editing; Andrew Zelhof, Funding acquisition, Writing – review and editing; Stephanie E Mohr, Norbert Perrimon, Supervision, Funding acquisition, Writing – review and editing; Amanda Simcox, Conceptualization, Formal analysis, Supervision, Funding acquisition, Investigation, Methodology, Writing - original draft, Project administration, Writing – review and editing

### Author ORCIDs

Daniel Mariyappa ![orcid] http://orcid.org/0000-0003-4775-1656
Molly Josifov ![orcid] http://orcid.org/0000-0002-2899-7186
Stephanie E Mohr ![orcid] http://orcid.org/0000-0001-9639-7708
Norbert Perrimon ![orcid] http://orcid.org/0000-0001-7542-472X
Amanda Simcox ![orcid] http://orcid.org/0000-0002-5572-7042

### Decision letter and Author response

Decision letter https://doi.org/10.7554/eLife.85814.sa1
Author response https://doi.org/10.7554/eLife.85814.sa2

## Additional files

### Supplementary files

• Supplementary file 1. Primary cultures and continuous lines produced from indicated genotypes. Glial primary cultures grew well at first and could be passaged several times; however, only one continuous line was produced. This line was cloned using single-cell dilution to produce three clonal derivatives. **Tracheal** lines were produced readily. Cloning the parental lines was not successful with either single-cell dilution or puro selection. **Mesodermal** lines were produced using expression of Ras$^{V12}$ with 24B-Gal4 but not Mef2-Gal4. Cloning of the continuous lines was done using single-cell dilution. **Neuronal**. Expression of Ras$^{V12}$ with neuronal Gal4 drivers (elav-Gal4 or scratch-Gal4) did not give rise to continuous lines. Cloning of lines generated by broad expression of Ras$^{V12}$ with

Act5C-GeneSwitch-Gal4 produced two clonal lines with **neuronal** characteristics and one with **hemocyte** characteristics. These were cloned using puro selection.

• Supplementary file 2. Analysis of marker gene expression in parental lines and clones. Cell lines and their clonal derivatives were stained with antibodies against the indicated markers. The fraction of cells staining positive was determined. The intensity and cellular location of the signal are indicated in cases when there was variation. The clones and parental lines highlighted were analyzed by RNAseq.

• Supplementary file 3. Fragments Per Kilobase of transcript per Million mapped reads (FPKM). FPKM values are shown for each of the clones and parental lines that were analyzed by RNAseq.

• MDAR checklist

### Data availability

Sequencing data have been deposited in GEO under accession code GSE219105.

The following dataset was generated:

| Author(s) | Year | Dataset title | Dataset URL | Database and Identifier |
|---|---|---|---|---|
| Mariyappa D, Luhur A, Zelhof A, Hu Y, Simcox A | 2022 | Continuous muscle, glial, epithelial, neuronal, and hemocyte cell lines for *Drosophila* research | https://www.ncbi.nlm.nih.gov/geo/query/acc.cgi?acc=GSE219105 | NCBI Gene Expression Omnibus, GSE219105 |

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

# Appendix 1

## Appendix 1—key resources table

| Reagent type (species) or resource | Designation | Source or reference | Identifiers | Additional information |
|---|---|---|---|---|
| Genetic reagent (*D. melanogaster*) | 24B/how-Gal4 | Bloomington *Drosophila* Stock Center | Stock # 1767; FLYB:FBti0150063; RRID:BDSC_1767 | FlyBase symbol: P(w[+mW.hs]=GawB) how[24B] |
| Genetic reagent (*D. melanogaster*) | repo-Gal4 | Bloomington *Drosophila* Stock Center | Stock # 7415; FLYB:FBti0018692; RRID:BDSC_7415 | FlyBase symbol: P(GAL4)repo |
| Genetic reagent (*D. melanogaster*) | btl-Gal4 | Bloomington *Drosophila* Stock Center | Stock # 78328; FLYB:FBti019793; RRID:BDSC_78328 | FlyBase symbol: P(GAL4-btl.S)3–2 |
| Genetic reagent (*D. melanogaster*) | Act5C-GeneSwitch-Gal4 | Bloomington *Drosophila* Stock Center | Stock # 9431; FLYB:FBti0003040,FBti0076553; RRID:BDSC_9431 | FlyBase symbol: P(UAS-GFP.S65T) Myo31DF[T2]; P(Act5C(-FRT)GAL4.Switch.PR)3 |
| Genetic reagent (*D. melanogaster*) | UAS-Ras$^{V12}$ (3) | Bloomington *Drosophila* Stock Center | Stock # 64195; FLYB:FBti0012505; RRID:BDSC_64195 | FlyBase symbol: P(w[+mC]=UAS-Ras85D.V12)TL1 |
| Genetic reagent (*D. melanogaster*) | UAS-Ras$^{V12}$ (2) | Bloomington *Drosophila* Stock Center | Stock # 64196; FLYB:FBti0180323; RRID:BDSC_64196 | FlyBase symbol: P(w[+mC]=UAS-Ras85D.V12)2 |
| Genetic reagent (*D. melanogaster*) | UAS-Ras$^{V12}$ with RMCE site (3) | Bloomington *Drosophila* Stock Center | Stock # 64197; FLYB: FBti0012505, FBti0102080; RRID:BDSC_64197 | FlyBase symbol: P(w[+mC]=UAS-Ras85D.V12)TL1, P(w[+mC]=attP.w[+].attP)JB89B |
| Genetic reagent (*D. melanogaster*) | UAS-GFP nuclear | Bloomington *Drosophila* Stock Center | Stock # 4775; FLYB: FBti0012492; RRID:BDSC_4775 | FlyBase symbol: P(UAS-GFP.nls)14 |
| Genetic reagent (*D. melanogaster*) | brat$^{dsRNA}$ | Bloomington *Drosophila* Stock Center | Stock # 34646; FLYB:FBti0140815; RRID:BDSC_34646 | FlyBase symbol: P(y[+t7.7] v[+t1.8]=TRiP.HMS01121)attP2 |
| Genetic reagent (*D. melanogaster*) | UAS-p35 baculovirus death inhibitor | Bloomington *Drosophila* Stock Center | Stock # 5072; FLYB:FBti0012594; RRID:BDSC_5072 | FlyBase symbol: P(w[+mC]=UAS-p35.H)BH1 |
| Genetic reagent (*D. melanogaster*) | Gal80ts | Bloomington *Drosophila* Stock Center | Stock # 7019; FLYB:FBti0027796; RRID:BDSC_7019 | FlyBase symbol: P(w[+mC]=tubP-GAL80[ts])20 |
| Cell line (*D. melanogaster*) | S2 | *Drosophila* Genomics Resource Center | Stock # 181; FLYB:FBtc0000181; RRID:CVCL_Z992 | Cell line maintained in N. Perrimon lab; FlyBase symbol: S2-DRSC. |
| Cell line (*D. melanogaster*) | 24B5-B8 | *Drosophila* Genomics Resource Center | Stock # 323; RRID:CVCL_C7G6 | 24B>Ras attP-L5-CloneB8 |
| Cell line (*D. melanogaster*) | 24BG1-G1 | *Drosophila* Genomics Resource Center | Stock # 324; RRID:CVCL_C7G7 | 24B>Ras attP-G1-CloneG1 |
| Cell line (*D. melanogaster*) | 24BG1-F3 | *Drosophila* Genomics Resource Center | Stock # 325; RRID:CVCL_C7G8 | 24B>Ras attP-G1-CloneF3 |
| Cell line (*D. melanogaster*) | Rbr6-2 | *Drosophila* Genomics Resource Center | Stock # 326; RRID:CVCL_C7G9 | repo>Ras bratdsRNA-L6-Clone2 |
| Cell line (*D. melanogaster*) | Rbr6-4 | *Drosophila* Genomics Resource Center | Stock # 327; RRID:CVCL_C7GA | repo>Ras bratdsRNA-L6-Clone4 |
| Cell line (*D. melanogaster*) | Rbr6-F9 | *Drosophila* Genomics Resource Center | Stock # 328; RRID:CVCL_C7GB | repo>Ras bratdsRNA-L6-CloneF9 |
| Cell line (*D. melanogaster*) | ActGSI-2 | *Drosophila* Genomics Resource Center | Stock # 329; RRID:CVCL_C7GC | Act5C-GS>Ras attP-LB-Clone6 |
| Cell line (*D. melanogaster*) | ActGSI-2 | *Drosophila* Genomics Resource Center | Stock # 330; RRID:CVCL_C7GD | Act5C-GS>Ras attP-GFP-LI-Clone2 |
| Cell line (*D. melanogaster*) | ActGSI-3 | *Drosophila* Genomics Resource Center | Stock # 331; RRID:CVCL_C7GE | Act5C-GS>Ras attP-GFP-LI-Clone3 |
| Cell line (*D. melanogaster*) | Btl3 | *Drosophila* Genomics Resource Center | Stock # 332; RRID:CVCL_B3N7 | btl>Ras attP-L3 |

*Appendix 1 Continued on next page*

*Appendix 1 Continued*

| Reagent type (species) or resource | Designation | Source or reference | Identifiers | Additional information |
|---|---|---|---|---|
| Cell line (*D. melanogaster*) | OK6-3 | *Drosophila* Genomics Resource Center | Stock # 281; RRID:CVCL_XF56 | OK6>Ras attP-L3 |
| Cell line (*D. melanogaster*) | Rbr6 | *Drosophila* Genomics Resource Center | Stock # 282; RRID:CVCL_XF57 | repo>Ras bratdsRNA-L6 |
| Cell line (*D. melanogaster*) | 24BG1 | *Drosophila* Genomics Resource Center | Stock # 283; RRID:CVCL_XF51 | 24B>Ras attP GFP-L1 |
| Cell line (*D. melanogaster*) | 24B5 | *Drosophila* Genomics Resource Center | Stock # 284; RRID:CVCL_XF52 | 24B>Ras attP-L5 |
| Cell line (*D. melanogaster*) | Btl7 | *Drosophila* Genomics Resource Center | Stock # 285; RRID:CVCL_XF53 | btl>Ras attP-L7 |
| Cell line (*D. melanogaster*) | Btl8 | *Drosophila* Genomics Resource Center | Stock # 286; RRID:CVCL_XF54 | btl>Ras attP-L8 |
| Cell line (*D. melanogaster*) | OK6-2 | *Drosophila* Genomics Resource Center | Stock # 287; RRID:CVCL_XF55 | OK6>Ras attP-L2 |
| cell line (*E. coli*) | DH5-alpha | Thermo Fisher | Cat. # 18265017 | Subcloning efficiency DH5-alpha competent cells |
| Transfected construct (*D. melanogaster*) | pAc5.1B-EGFP | Addgene | Cat. # 21181; http://n2t.net/addgene: 21181; RRID:Addgene_21181 | pAc5.1B-EGFP was a gift from Elisa Izaurralde |
| Transfected construct (*D. melanogaster*) | pCoPURO | Addgene | Cat. # 17533; http://n2t.net/addgene: 17533; RRID:Addgene_17533 | pCoPURO was a gift from Francis Castellino |
| Antibody | AffiniPure Rabbit Anti-Horseradish Peroxidase (Rabbit polyclonal) | Jackson ImmunoResearch | Cat. # 323-005-021; RRID: AB_2314648 | Rabbit polyclonal; IF (1:500) |
| Antibody | 22C10 (mouse monoclonal) | Developmental Studies Hybridoma Bank | Cat. # 22C10 RRID: AB_528403. FBgn0259108 | 22C10 was deposited to the DSHB by Benzer, S./Colley, N.; mouse monoclonal; IF (1:100) |
| Antibody | Rat-Elav-7E8A10 anti-elav (rat monoclonal) | Developmental Studies Hybridoma Bank | Cat. # Rat-Elav-7E8A10 anti-elav, RRID:AB_528218 | Rat-Elav-7E8A10 anti-elav was deposited to the DSHB by Rubin, G.M.; rat monoclonal; IF (1:100) |
| Antibody | 8D12 anti-Repo (mouse monoclonal) | Developmental Studies Hybridoma Bank | Cat. # 8D12 anti-Repo, RRID:AB_528448 | 8D12 anti-Repo was deposited to the DSHB by Goodman, C.; mouse monoclonal; IF (1:100) |
| Antibody | 1D4 anti-Fasciclin II (mouse monoclonal) | Developmental Studies Hybridoma Bank | Cat. # 1D4 anti-Fasciclin II, RRID:AB_528235 | 1D4 anti-Fasciclin II was deposited to the DSHB by Goodman, C.; mouse monoclonal; IF (1:100) |
| Antibody | Guinea pig anti-Twist (guinea pig polyclonal) | M.Levine, UC Berkeley, CA | | A gift from M. Levine, UC Berkeley, CA; guinea pig polyclonal; IF (1:500) |
| Antibody | 3E8-3D3 (mouse monoclonal) | Developmental Studies Hybridoma Bank | Cat. # 3E8-3D3, RRID:AB_2721944 | 3E8-3D3 was deposited to the DSHB by Saide, J.D.; mouse monoclonal; IF (1:100) |
| Antibody | DCAD2 (rat monoclonal) | Developmental Studies Hybridoma Bank | Cat. # DCAD2, RRID:AB_528120 | DCAD2 was deposited to the DSHB by Uemura, T.; rat, monoclonal; IF (1:100) |
| Antibody | Rabbit anti-DMef2 (rabbit polyclonal) | doi:10.1101/gad.9.6.730 | | A gift from J. R. Jacobs; rabbit polyclonal; IF (1:500) |
| Antibody | Mouse anti-H2 (mouse monoclonal) | doi:10.1073/pnas.0436940100 | | *Kurucz et al., 2003*; IF (1:10) |
| Antibody | Cy3 AffiniPure Goat Anti-Mouse IgG (H+L) (Goat polyclonal) | Jackson ImmunoResearch | Cat. # 115-165-003; RRID: AB_2338680 | Goat polyclonal; IF (1:1000) |
| Antibody | Cy3 AffiniPure Goat Anti-Rat IgG (H+L) (Goat polyclonal) | Jackson ImmunoResearch | Cat. # 112-165-003; RRID: AB_2338240 | Goat polyclonal; IF (1:1000) |

*Appendix 1 Continued on next page*

*Appendix 1 Continued*

| Reagent type (species) or resource | Designation | Source or reference | Identifiers | Additional information |
|---|---|---|---|---|
| Antibody | Cy3 AffiniPure Goat Anti-Guinea Pig IgG (H+L) (Goat polyclonal) | Jackson ImmunoResearch | Cat. # 106-165-003; RRID: AB_2337423 | Goat polyclonal; IF (1:1000) |
| Antibody | Cy3 AffiniPure Goat Anti-Rabbit IgG (H+L) (Goat polyclonal) | Jackson ImmunoResearch | Cat. # 111-165-045; RRID: AB_2338003 | Goat polyclonal; IF (1:1000) |
| Antibody | Donkey anti-Rabbit IgG (H+L) Highly Cross-Adsorbed Secondary Antibody, Alexa Fluor 488 (donkey polyclonal) | Thermo Fisher | Cat. # A-21206; RRID: AB_2535792 | Donkey polyclonal; IF (1:1000) |
| Commercial assay or kit | Effectene Transfection Reagent | QIAGEN | Cat. # 301425 | |
| Commercial assay or kit | NucleoSpin Plasmid Kit (No Lid) | Macherey-Nagel | Cat. # 740499.250 | |
| Commercial assay or kit | DNeasy Blood & Tissue Kit | QIAGEN | Cat. # 69504 | |
| Chemical compound, drug | KaryoMAX Colcemid Solution in PBS | Gibco Thermo Fisher | Cat. # 15212–012 | |
| Chemical compound, drug | Schneider's Insect Medium | Sigma-Aldrich | Cat. # S0146 | |
| Chemical compound, drug | FBS | Gibco Thermo Fisher | Cat. # 26140–079 | |
| Chemical compound, drug | 0.05% Trypsin–EDTA (1×) | Gibco Thermo Fisher | Cat. # 25300–120 | |
| Chemical compound, drug | Penicillin–streptomycin (10,000 U/ml) | Gibco Thermo Fisher | Cat. # 15140122 | |
| Chemical compound, drug | Mifepristone | Invitrogen Thermo Fisher | Cat. # H11001 | |
| Chemical compound, drug | 20-Hydroxyecdysone | Sigma-Aldrich | Cat. # H5142 | |
| Chemical compound, drug | VECTASHIELD Antifade Mounting Medium With DAPI | Vector Laboratories | Cat. # H1200 | |
| Software, algorithm | GraphPad Prism version 9.5.1 | https://www.graphpad.com/ | RRID:SCR_002798 | |
| Software, algorithm | Fiji | doi:10.1038/nmeth.2019 | RRID:SCR_002285 | |

