## [Editor Report]

This valuable work describes the establishment and characterization of new cell lines derived from specific tissues of the fruit fly *Drosophila*. The evidence supporting the claims of the authors is convincing, with rigorous characterization of the cell lines and incorporation of their transcriptomes into *Drosophila* Gene Expression Tool website for user-friendly access. These lines will be a valuable resource that complements in vivo *Drosophila* genetics, improving biochemistry and facilitating high-throughput screening.

---

## [Decision Letter]

**Decision letter after peer review:**

Thank you for submitting your article "Continuous muscle, glial, epithelial, neuronal, and hemocyte cell lines for *Drosophila* research" for consideration by *eLife*. Your article has been reviewed by 3 peer reviewers, including Erika A Bach as the Reviewing Editor and Reviewer #1, and the evaluation has been overseen by Claude Desplan as the Senior Editor. The following individual involved in the review of your submission has agreed to reveal their identity: David Bilder (Reviewer #2).

Essential revisions:

1. Increase the 'identity' of the cell lines

– Provide information and possibly new data to increase the utility of Table 2

– Provide information on what expected marker genes are NOT expressed in the new cell lines

– Compare the new lines to existing lines, especially the workhorse S2 cells.

– Provide information/commentary on growing in new cells line in large numbers or employing them in high-throughput screening.

2. Improve user access to genomics data. Downloading and accessing GEO files are not user-friendly. Genomic data should be accessible to the potential users of these cell lines.

– Provide tabular genome-wide data for each cell line.

– Consider generating a website with these data

3. Improve the images and figure legends

– Provide higher magnification views to support the conclusions.

– Figure 2: show Ecad staining on a non-epithelial cell line with comparable density.

– Figure 3D: show more clearly the multinucleate cells.

– Provide information on magnification for images.

4. Methods

– Mention how specific the GAL4 drivers are in embryos (possibly by providing real time expression, maybe using G-trace).

– Provide a detailed description of how the new cell lines were generated (rather than cite Simcox, 2013).

– Provide a detailed description of the culture conditions for the new cell lines.

– Provide reagent tables and RRIDs for all reagents

5. Improve karyotyping.

*Reviewer #1 (Recommendations for the authors):*

1. The authors should review their reference manager because not all of the references are cited in the same way. Most are last name comma first initial but A. Simcox is frequently cited.

2. Under the Methods section, the authors should provide a description of how they generated the cell lines (rather than cite Simcox, 2013).

3. Is it possible to put scale bars on the micrographs? If not, could the authors please state the magnification?

*Reviewer #2 (Recommendations for the authors):*

1) The critical assignment of the 'identity' of the cell lines would benefit from more detail and explanation. The process is a bit confusing. For instance, in Table 2, why is the dataset Whole body for some and a specific organ for others? In the latter case, why does Btl8 map to the oenocyte sample? On line 182, the authors say that the glial lines did not result in as clear a similarity as other lines, but that is not obvious from the values in Table 2. Is it possible/useful to compare the enrichment scores of Table 2 to those of well-characterized tissue-specific mammalian cell lines and their in vivo equivalents (say MDCK cells and renal tubule cells) for perspective?

2). Related, the paper would benefit from an explicit critical discussion of the cell lines. For instance, what expected marker genes are NOT expressed? E.g. the tracheal-derived cell lines L8 do not express trh, L7 downregulates trh. In what ways are these not simple immortalized cells of in vivo equivalents -that may be important for people thinking about using them? The authors' simple statements e.g. line 212 do a disservice to the nuances of identity.

3) The authors could make clear how these new lines compare to existing lines. For instance, S2 was generally thought to be hemocyte-like, how does it compare to Act GSI-3? What reason would researchers interested in hemocyte biology want to use one over the other? An example of this value is the comparison of transfection efficiency to S2 (line 237).

4) Any experience with growing in large numbers, or in high-throughput screening format? Since these are promoted as uses, it would be very helpful to have a perspective from the authors, who include those most experienced with the variety of existing cell lines and their use in these contexts.

Other points – not absolutely needed for accceptance

In general, the figures are rather sparse, there is room for more data such as those suggested above.

Any speculation as to why so many of the lines correlate with an adult, rather than embryonic, profiles when the tissue source was embryos?

Figure 2: would be nice to show Ecad staining on a non-epithelial cell line with comparable density.

Would be nice to mention how specific the GAL4 drivers are in embryos

The culture conditions for the cell lines, which is critical information, are buried in a section about 'RNA extraction'.

Figure 3D: would be nice to show more clearly the multinucleate cells.

For the lines where there was a failure to derive (e.g. *Mef2*; elav) -is it possible that the levels of active Ras were too high to be compatible with life and/or immortalization?

The new lines all group more closely with each other in Figure S1 than with any previous line, regardless of the supposed tissue represented. Presumably, this is due to an activated Ras expression signature. Or is their contribution from their more recent time of derivation, or their more similar genetic background? If you 'filter out' the generic Ras expression signature, does that help with the assignment of the cell type?

Discussion about the role of continuous Ras expression would be nice. Inspection of the figure legends 4C and 5C suggests that turning off GAL4 or Ras activity makes the cells quiescent, but explicit information about this, how long they can live in this state, whether there are obvious effects on differentiation, etc. would be useful.

*Reviewer #3 (Recommendations for the authors):*

The paper is clear and well-written, but I believe that the work needs to be extended to make it useful.

1) The RNAseq characterization is perhaps the most important and global aspect of the utility, because someone interested in a particular gene, could check to see if it was well expressed and in which of the novel (or existing) cell lines. I did not see tabular genome-wide data that would fit this bill. The expression data is basically not presented in the manuscript. The authors are used to presenting data to the community through the DGRC, DRSC, and FlyBase. They need to make these data easy to access. It is hard to judge the GEO submission as it is still listed as private.

2) The phase images are not sufficient to understand the cell biology nature of these cell lines. High magnification views that show higher content information on the cells are needed. It is nice to show the hormonally induced differentiation onset using one or a few markers, but for a resource, I would hope for more granularity. Again it is difficult to make choices of cell lines for experiments with this resolution.

3) Although I do not recall reading it explicitly, I assume that the chromosome spreads were scored based on chromosome size. This is not very satisfying in the era of chromosome paints and especially DNAseq (e.g. PMID: 25262759).

4) The figure legends should indicate what is being detected. "Expressed" is too vague.

5) Reagent tables are much easier to read than in-text methods with names and undescribed identifiers.

---

## [Author Response]

Essential revisions:1. Increase the 'identity' of the cell lines– Provide information and possibly new data to increase the utility of Table 2– Provide information on what expected marker genes are NOT expressed in the new cell lines– Compare the new lines to existing lines, especially the workhorse S2 cells.– Provide information/commentary on growing in new cells line in large numbers or employing them in high-throughput screening.

In Table 2, we have added the cell-type based on in situ data. This complements the single cell RNAseq and confirms the cell type identity.

We have added a section on transcription factors, not expressed in the new cell lines, which might be expected for that cell type. Also, we note that the RNAseq data is for undifferentiated cells and genes not expressed in the proliferative state may be expressed in differentiated cells. This is one reason why information on what is not expressed may not be definitive.

We have included figures that show the morphology of the cells, including S2 cells (Figures 1 and 8). These show that none of the new lines are similar to S2, including the hemocyte-like cell line ActGSI-3. While both S2 and ActGSI-3 cells are round and grow in suspension, S2 cells grow individually, whereas ActGSI-3 form large cell rafts.

In Table 3, we have added data about the confluent density of the cells. These numbers show that the cells can be grown to densities of 1.4-8.1 x 10^6^ in a single well of a 12 well plate. In this experiment S2 cells grew to a confluent density of 6.2 X 10^6^. As all cells, except ActGSI-3, are adherent, the cells are also potentially well suited for multi-well plate assays.

2. Improve user access to genomics data. Downloading and accessing GEO files are not user-friendly. Genomic data should be accessible to the potential users of these cell lines.– Provide tabular genome-wide data for each cell line.– Consider generating a website with these data

We added a supplementary table (Table S3) with genome-wide expression levels for each gene in each cell line (Fragments Per Kilobase per Million mapped fragments, FPKM).

The dataset has been imported into the *Drosophila* Gene Expression Tool (DGET) database (https://www.flyrnai.org/tools/dget/web/), which is the bulk RNAseq data portal at *Drosophila* RNAi Screening Center (DRSC). DGET can be used to query the expression levels for the genes of interest as well as search the genes with similar expression pattern for any input gene. The expression dataset is called “Ras cell lines”.

3. Improve the images and figure legends– Provide higher magnification views to support the conclusions.– Figure 2: show Ecad staining on a non-epithelial cell line with comparable density.– Figure 3D: show more clearly the multinucleate cells.– Provide information on magnification for images.

The new Figure 1 shows the cells at high magnification. This was a very useful suggestion because it allows comparison of cell morphologies within and between lineages. For example, the glial cells have a long spindle-like shape, and the epithelial cells are cuboidal.

Figure 6 shows Btl3 and S2 cells stained for the epithelial marker E-Cadherin/Shotgun. Btl3 cells expressed Ecad at the cell periphery and S2 cells had low level cytoplasmic expression.

New panels in Figure 5 show the multinucleate muscle cells.

Scale bars are included in Figures 1 and 8, which show phase images of the cells.

4. Methods– Mention how specific the GAL4 drivers are in embryos (possibly by providing real time expression, maybe using G-trace).– Provide a detailed description of how the new cell lines were generated (rather than cite Simcox, 2013).– Provide a detailed description of the culture conditions for the new cell lines.– Provide reagent tables and RRIDs for all reagents

We find that additional factors other than specificity of expression of Gal4 are necessary for successful cell line development. This is illustrated for the muscle lineage where we tested two well characterized drivers—*Mef2*-Gal4 and 24B/how-Gal4. Only 24B/how-Gal4 produced continuous cell lines. In a new figure we show that Ras^V12^ expression with *Mef2*-Gal4 in muscle cells disrupted muscle development and that in primary cultures the Ras^V12^-expressing cells (marked with GFP) failed to attach and grow (Figure 5—figure supplement 4). We. Do not the know the basis for this; however, expression level is a possibility. Even after successful cell line generation it is important to use a battery of assays to determine lineage characteristics because cells change as they adapt for survival in vitro. This is exemplified by some of the 24B-Gal4 derived lines that failed to differentiate after extended passages.

A detailed primary culture procedure is provided in the Methods section.

Culture conditions and passaging regimes for the new cell lines are provided in the Methods section. This method also refers to Figure 8, which shows the cells at confluence and will help users determine when cells are ready for sub-culturing (passaging).

All reagents and RRIDs (when available) are provided.

5. Improve karyotyping.

We redid the karyotypes for all the cell lines and include karyograms for each. The quality is greatly improved and equivalent to the standard for the field. The figures showing the karyotypes and the chromosomes ordered in a karyogram appear in supplemental figures for each cell line and the refined karyotypes are summarized in the revised Table 3.